# Glacier Algae accelerate melt rates on the south-western Greenland Ice Sheet

Joseph M. Cook[1], Andrew J Tedstone[2], Christopher Williamson[3], Jenine McCutcheon[4], Andrew J. Hodson[5,6], Archana Dayal[1,5], McKenzie Skiles[7], Stefan Hofer[2], Robert Bryant[1], Owen McAree[8], Andrew McGonigle[1], Jonathan Ryan[11], Alexandre M. Anesio[12], Tristram D.L. Irvine-Fynn[10], Alun Hubbard[13], Edward Hanna[14], Mark Flanner[15], Sathish Mayanna[16], Liane G. Benning[16,17,4], Dirk van As[18], Marian Yallop[3], James B. McQuaid[4], Martyn Tranter[2,], Thomas Gribbin[2].

[1] Department of Geography, University of Sheffield

[2] Center for Glaciology, University of Bristol

[3] Department of Biosciences, University of Bristol

[4] School of Earth and Environment, University of Leeds

[5] Department of Geology, University Centre in Svalbard

[6] Department of Environmental Sciences, Western Norway University of Applied Sciences, 6856 Sogndal, Norway

[7] Department of Geography, University of Utah

[8] Faculty of Science, Liverpool John Moores University

[9] School of Geociences, University of Sydney, Sydney, NSW 2006, Australia

[10] Department of Geography and Earth Science, Aberystwyth University, Wales, SY23 3DB, UK

[11] Institute at Brown for Environment and Society, Brown University, Providence, Rhode Island, USA

[12] Department of Environmental Science, Aarhus University, 4000 Roskilde, Denmark

[13] Centre for Gas Hydrates and Climate, University of Tromso, Norway

[14] School of Geography and Lincoln Centre for Water and Planetary Health, University of Lincoln

[15] University of Michigan, Ann Arbor, Michigan, USA

[16] German Research Centre for Geosciences, GFZ, Potsdam, Germany

[17] Department of Earth Sciences, Free University of Berlin, Germany

[18] Geological Survey of Denmark and Greenland, Copenhagen, Denmark

*Correspondence to*: Joseph Cook (joe.cook@sheffield.ac.uk)

**Abstract.** Melting of the Greenland Ice Sheet (GrIS) is the largest single contributor to eustatic sea level and it is amplified by the growth of pigmented algae on the ice surface that increase solar radiation absorption. This biological albedo-reducing effect and its impact upon sea level rise has not previously been quantified. Here, we combine field spectroscopy with a radiative transfer model, supervised classification of UAV and satellite remote-sensing data and runoff modelling to calculate biologically-driven ice surface ablation. We

demonstrate that algal growth led to an additional 4.4 – 6.0 Gt of runoff from bare ice in the south-western sector of the GrIS in summer 2017, representing 10 – 13 % of the total. In localised patches with high-biomass accumulation, algae accelerated melting by up to 26.15 ± 3.77 % (standard error). 2017 was a high albedo year, so we also extended our analysis to the particularly low-albedo 2016 melt season. The runoff from the south-western bare-ice zone attributed to algae was much higher in 2016, at 7.3 – 10.9 Gt, although the proportion of the total runoff contributed by algae was similar at 9 – 13%. Across a 10,000 km$^2$ area around our field site, algae covered similar proportions of the exposed bare ice zone in both years (57.99 % in 2016, 58.89 % in 2017), but more of the algal ice was classed as "high-biomass" in 2016 (8.35 %) than 2017 (2.54 %). This interannual comparison demonstrates a positive feedback where more widespread, higher biomass algal blooms are expected to form in high melt years where the winter snowpack retreats further, earlier, providing a larger area for bloom development and also enhancing the provision of nutrients and liquid water liberated from melting ice. Our analysis confirms the importance of this biological albedo feedback and that its omission from predictive models leads to the systematic underestimation of Greenland's future sea level contribution, especially because both the bare-ice zones available for algal colonization and the length of the biological growth season are set to expand in the future.

## 1 Introduction

Mass loss from the Greenland Ice Sheet (GrIS) has increased over the past two decades (Shepherd et al., 2012; Hanna et al., 2013) and is the largest single contributor to cryospheric sea level rise, adding 37% or 0.69 mm yr$^{-1}$ between 2012-2016 (Bamber et al., 2018). This is due to enhanced surface melting (Ngheim et al., 2012) that exceeds calving losses at the ice-sheet's marine-terminating margins (Enderlin et al., 2014; van den Broeke et al., 2016). Surface melting is controlled by net solar radiation, which in turn depends upon the albedo of the ice surface, making albedo a critical factor for modulating ice-sheet mass loss (Box et al., 2012; Ryan et al. 2018a). The largest shift in albedo occurs when the winter snow retreats to expose bare glacier ice. However, there are several linked mechanisms that then change the albedo of the exposed ice and determine its rate of melting, including meltwater accumulation, ice surface weathering and the accumulation of light-absorbing particles (LAPs), such as soot (Flanner et al., 2007) and mineral dust (Skiles et al., 2017). Photosynthetic algae also reduce the albedo of the GrIS (Uetake et al., 2010; Yallop et al., 2012; Stibal et al., 2017; Ryan et al., 2017, 2018b). Despite being identified in the late 1800's (Nordenskiöld, 1875) their effects have not yet been quantified, mapped or incorporated into any predictive surface mass balance models (Langen et al. 2017; Noel et al., 2016; Fettweis et al. 2017). Hence, biological growth may play an important yet under-appreciated role in the melting of the Greenland Ice Sheet and its contributions to sea level rise (Benning et al., 2014).

The snow-free surface of the GrIS has a conspicuous dark stripe along its western margin which expands and contracts seasonally, covering 4 - 10% of the ablating bare-ice area (Shimada et al., 2016). The extent and

darkness of this "Dark Zone" may be biologically and/or geologically controlled (Wientjes et al., 2011; 2016;
Tedstone et al., 2017; Stibal et al., 2017). There is a growing literature demonstrating the albedo-reducing role played by a community of algae that grow on glacier ice on the eastern (Lutz et al. 2014) and western (Uetake et al. 2010; Yallop et al. 2012; Stibal et al. 2017; Tedstone et al. 2017; Williamson et al. 2018) GrIS. The algal community on the GrIS is dominated by *Mesotaenium berggrenii*, and *Ancylonema nordenskioldii* (Yallop et al., 2012; Stibal et al., 2017; Williamson et al., 2018; Lutz et al. 2018; Williamson et al. 2019),
which are collectively known as "glacier algae" to distinguish them from snow algae and sea-ice-algae. The presence of these glacier algae reduces the albedo of the ice surface, mostly due to a brown-purple purpurogallin-like pigment (Williamson et al., 2018; Stibal et al., 2017; Remias et al., 2012).

An equivalent albedo reduction due to algae has also been studied on snow. Worldwide, snow algal
communities are dominated by unicellular *Chlamydomonaceae*, the most abundant of which belong to the collective taxon *Chlamydomonas nivalis* (Leya et al. 2004). These algae have been shown to be associated with low-albedo snow in Eastern Greenland (Lutz et al. 2014) and to be responsible for 17 % of snowmelt in Alaska (Ganey et al. 2017). However, for glacier algae, quantification of the biological albedo reduction, radiative forcing and melt acceleration has remained elusive due to the difficulty of separating biological
from non-biological albedo-reducing processes and a lack of diagnostic biosignatures for remote-sensing. For snow, remote detection has been achieved by measuring the "uniquely biological" chlorophyll absorption feature at 680 nm (Painter et al. 2001), a broader carotenoid absorption feature (Takeuchi et al. 2006), a normalised difference spectral index (Ganey et al. 2017) and a spectral unmixing model (Huovinen et al. 2018). However, these signature spectra can be ambiguous for glacier algae due to the presence of the
phenolic pigment with a broad range of absorption across the UV and VIS wavelengths that obscures features associated with other pigments in raw reflectance spectra, and is further complicated by the highly variable optics of the underlying ice and mixing of algae with other impurities.

The Dark Zone is of the order $10^5 \, \text{km}^2$ in extent and is undergoing long-term expansion (Shimada et al.,
2016; Tedstone et al., 2017). Quantifying the impact of algal colonization on the Dark Zone is therefore paramount. Upscaling of unmanned aerial vehicle (UAV) observations made in a small sector of the Dark Zone to satellite data has demonstrated that "distributed impurities" including algae exert a primary control on the surface albedo, but isolating the biological effect and upscaling to the regional scale has been prevented by a lack of spectral resolution and ground validation (Ryan et al., 2018a). Recently, Wang et al.
(2018) have applied the vegetation red-edge (difference in reflectance between 673 and 709 nm) to map glacier algae over the south-western GrIS. However, the red-edge is potentially vulnerable to false positives due to red mineral dusts (Seager et al. 2005; Cook et al. 2017b) which was not discounted using ground observations in their study. Furthermore, their remote-sensing used Sentinel-3 OLCI data at a coarse ground resolution (300 m) – introducing high uncertainty due to sub-pixel surface heterogeneity - and did not
account for the highly variable optics of the underlying ice, meaning there is likely large uncertainty

associated with their predictions of cell abundance. Neither of these previous studies quantified the effect of glacier algae effect on albedo or melt at the regional scale.

Here, we directly address these issues, resolving a major knowledge gap limiting our ability to forecast ice-sheet melt rates into the future. First, we use spectroscopy to quantify the effect of algae on albedo and radiative forcing in ice. We then use a new radiative transfer model to isolate the effects of individual light-absorbing particles on the ice surface for the first time, enabling a comparison between local mineral dust and algae and providing the first candidate albedo parameterisation that could enable glacier algae to be incorporated into mass balance models. To determine spatial coverage, we apply a supervised-classification algorithm (random-forest) to map glacier algae in multispectral UAV and satellite data. Runoff modelling informed by our empirical measurements and remote-sensing observations enables us to estimate the biological contribution to GrIS runoff for the first time.

## 2. Field sites and Methods

### 2.1 Overview

In this study we present a suite of empirical, theoretical and remote-sensing data to quantify and map algal contributions to melting on the south-western GrIS. At our field site we paired spectral reflectance and albedo measurements with removal of surface ice samples for biological and mineralogical analyses in order to quantify the relationship between cell abundance and broadband and spectral albedo. The imaginary part of the refractive index of the local mineral dusts and the purpurogallin-type phenolic compound that dominates absorption in the local glacier algae were measured in the laboratory and incorporated into a new radiative transfer model. The albedo effects of each impurity were thus examined in isolation and compared. At the same time, we also undertook a sensitivity study with other bulk dust optical properties from previous literature to further test the potential role of mineral dusts in darkening the ice surface. Furthermore, by combining albedo measurements with incoming irradiance spectra and measurements of local melt rates, we estimated the radiative forcing and the proportion of melting that could be attributed to algae in areas of high and low algal biomass ($H_{bio}$ and $L_{bio}$). At our field site we made UAV flights with a multispectral camera in order to map algal coverage at high spatial resolution. We achieved this by training a random-forest (RF) algorithm on our field spectroscopy data to classify the ice surface into discrete categories including $H_{bio}$ and $L_{bio.}$ This enabled estimates of algal coverage in a 200 x 200m area at our field site. We then retrained our classifier for Sentinel-2 satellite imagery and used this to upscale further within the south-western region of the GrIS. With these estimates of algal coverage from our remote-sensing imagery and calculations of the proportion of melting attributed to algae from our field data, we were able to estimate runoff attributed to algae using van As et al.'s (2017) runoff model. The details of each stage of our methodology are provided in the following sections 2.2 – 2.10.

## 2.2 Field Site

Experiments were carried out at the Black and Bloom Project field site (67.04 N, 49.07 W, Fig 1), near the Institute for Marine and Atmospheric research Utrecht (IMAU) Automatic Weather Station 'S6' on the south-western Greenland Ice Sheet between 10 – 22$^{nd}$ July 2017. We established a 200 x 200 m area for UAV mapping (centred on 49° 21' 0.50" W, 67° 4' 40.42" N) where only essential access was allowed (e.g. for placing ground control points, GCPs, for georectifying our UAV images) and sample removal was prohibited. We also delineated an additional adjacent 20 x 200 m area that we referred to as the "sampling strip" in which we made spectral reflectance and albedo measurements paired with removal of samples for biological and mineralogical analyses, as detailed in the following sections. The sampling strip was subdivided into smaller subregions that were then systematically visited each day of our field season. This was necessary because ice surface samples were destructively removed for analysis and this method ensured that each area visited had not been disturbed by our presence on previous days. Some ancillary directional reflectance measurements were also made at the same field site between 15 - 25$^{th}$ July 2016 and appended to our training dataset for supervised classification (Section 2.8).

## 2.3 Field Spectroscopy

At each site in our sampling strip, albedo was measured using an ASD (Analytical Spectral Devices, Colorado) Field-Spec Pro 3 spectroradiometer with ASD cosine collector. The cosine collector was mounted horizontally on a 1.5 m crossbar levelled on a tripod with a height between 30 - 50 cm above the ice surface. The cosine collector was positioned over a sample surface, connected to the spectroradiometer using an ASD fibre optic, then the spectroradiometer was controlled remotely from a laptop, meaning the operators could move away from the instrument to avoid shading it. Two upwards and two downwards looking measurements were made in close succession (~ 2 minutes) to account for any change in atmospheric conditions, although the measurements presented were all made during constant conditions of clear skies at solar noon ± 2 hours. Each retrieval was the average of > 20 replicates.

Immediately after making the albedo measurements, the cosine collector was replaced with a 10 degree collimating lens, enabling a nadir-view hemispheric conical reflectance factor (HCRF) measurement to be obtained. For HCRF measurements the upwards looking measurements were replaced with HCRF measurements of a flat Spectralon® panel with the spectroradiometer in reflectance mode. This protocol was followed for every sample surface with both albedo and directional reflectance measurements taking less than 5 minutes. We closely followed the methodology described by Cook et al., (2017b). Albedo is the most appropriate measurement for determining the surface energy balance, while the HCRF is closer to the measurements made by aerial remote-sensing and less sensitive to stray light reflecting from surfaces other than the homogenous patch directly beneath the sensor. We therefore used the albedo for energy balance calculations and the HCRF for remote sensing applications in this study.

## 2.4 Biological Measurements

Immediately following the albedo and HCRF measurements, ice from within the viewing area of the
spectrometer was removed using a sterile blade and scooped into sterile whirlpak bags, melted in the dark
and immediately fixed with 3% gluteraldehyde. The samples were then returned to the University of Bristol
and University of Sheffield where microscopic analyses were undertaken. Samples were vortexed thoroughly
before 20 µL was pipetted into a Fuchs-Rosenthal haemocytometer. The haemocytometer was divided into 4
x 4 image areas. These were used to count a minimum of 300 cells to ensure adequate representation of
species diversity (where possible – low abundance samples had as few as one cell per haemocytometer). The
volume of each image area was used to calculate cells per mL. Biovolume was determined by measuring the
long and short axes of at least ten cells from each species in each sample using the measure tool in the GNU
Image Manipulation Program (GIMP). The morphology of the cells in the images was used to separate them
into two species: *Mesotaenium bergrenii* and *Ancylonema nordenskioldii*. These dimensions were then used
to calculate the mean volume of each species in each sample, assuming the cells to be circular cylinders
(after Hillebrand, 1999 and Williamson et al., 2018). The average volume was multiplied by the number of
cells for each species and then summed to provide the total biovolume for each sample.

## 2.5 Mineral and algal optical properties and radiative transfer modelling

A new radiative transfer package, BioSNICAR_GO, was developed for this study and was used to predict the
albedo of snow and ice surfaces with algae and mineral dusts. We made a series of major updates and
adaptations to the BioSNICAR model presented by Cook et al. (2017b). The package is divided into a bio-
optical scheme wherein the optical properties of light-absorbing impurities and ice crystals can be calculated
using Mie scattering (for small spherical particles such as black carbon or snow) or geometrical optics (for
large and/or aspherical particles such as glacier algae, larger mineral dust particles and large ice crystals) and
a two-stream radiative transfer model based on SNICAR (Flanner et al. 2007) which incorporates the
equations of Toon et al. (1989). A schematic of the model structure is provided in Supp. Info 1.

To incorporate glacier algae into BioSNICAR_GO, geometrical optics was employed to determine the single
scattering optical properties of the glacier algae, since they are large ($\sim 10^3$ µm³: far outside the domain of
Mie scattering) and best approximated as circular cylinders (Hillebrand, 1999; Lee and Pilon, 2003). Our
approach is adapted from the geometric optics parameterisation of van Diedenhoven (2014). The inputs to
the geometric optics calculations are the cell dimensions and the complex refractive index. The imaginary
part of the refractive index was calculated using a mixing model based upon Cook et al., (2017b) where the
absolute mass of each pigment in the algal cells was measured in field samples. The absorption spectra for
the algal pigments is provided in Fig 2A. We updated Cook et al.'s (2017b) mixing model to apply a volume-
weighted average of the imaginary part of the refractive index of water and the algal pigments so that the

simulated cell looks like water at wavelengths where pigments are non-absorbing. We consider this to be more physically realistic than having cells that are completely non-absorbing at wavelengths > 0.75 µm, especially since a water fraction ($X_w$) is used in the calculations to represent the non-pigmented cellular components of the total cell volume. This approach also prevents the refractive index from becoming infinite when the water fraction is zero, removing the constraint $0 < X_w < 1$ from the bio-optical scheme in the original BioSNICAR model. Based upon experimental evidence in Dauchet et al (2015) for the model species *C. reinhardii*, the real part of the refractive index has been updated from 1.5 (in Cook et al. 2017b) to 1.4. The absorption coefficients from which the imaginary refractive index is calculated are from Dauchet et al (2015) apart from the purpurogallin-type phenol whose optical properties were determined empirically (Fig 2A). The calculated optical properties were added to the lookup library for BioSNICAR_GO for a range of cell dimensions. For the simulations presented in this study, we included two classes of glacier algae representing *Mesotenium Bergrenii* and *Ancylonema nordenskioldii* with length and diameter and also the relative abundance of each species matching the means measured in our microscopy described in Section 2.4. In simulations not shown here we found that ice albedo was relatively insensitive to the dimensions of the cells within a realistic range of lengths and diameters. For example, in a simulation with constant ice optics (snow of constant grain size 400 µm, density 400 kg m$^{-3}$ and snowpack thickness 50 mm) and a biomass mixing ratio of $10^5$ ppb in the upper 1 mm layer, changing the length of the algal cells from 10 to 40 um changed the albedo by less than 0.004. The albedo was more sensitive to cell diameter; with the same ice optics and biomass mixing ratio as described above and an algal cell 40 µm long, changing the cell diameter from 4 µm to 12 µm changed the surface albedo by 0.008. This low sensitivity to cell length and diameter is likely because all of the cells considered here are large from a radiative transfer perspective.

For mineral dusts, we measured the optical properties of local mineral dusts by first removing the organic matter from the impurties present in melted $H_{bio}$ ice and measuring the PSD using scanning electron microscopy (see Supp Info 2 for full sample preparation and PSD measurement details). We then arranged the mineral dust samples into an optically thick layer on a microscope slide and pressed them tightly against the open aperture of a Thorlabs IS200-4 2" integrating sphere. The other apertures were covered with SM05CP2C caps and the sample reflectance was measured using the same ASD Field Spec Pro 3 as was used for field measurements. The PSD and reflectance of the sample were then used to determine the complex refractive index of the bulk mineral dust mixture by inverting the Discrete Ordinates radiative transfer model (DISORT) following the methodology of Skiles et al. (2017). Briefly, the measured reflectance is used as a target for repeated runs of the DISORT model with varying refractive indices. The refractive index that gives the lowest root mean square error across the solar spectrum is determined to be the real refractive index of the bulk mixture. This is then used to forward model the optical properties of the mineral dust using Mie scattering theory. This was undertaken for mineral dust of radii 0.01 – 10 µm at a resolution of 0.01 µm. The measured PSD was then used to apply a weighted average to the estimated single scattering properties to provide a bulk refractive index for the measured mineral sample. This was then added

to the lookup library for the radiative transfer model. The new radiative transfer package BioSNICAR_GO was used to predict the albedo of snow and ice surfaces with our field-measured mineral dust.

Since we only had two samples from southwestern Greenland, we also incorporated into our radiative transfer model published optical properties for bulk minerals by Polashenski et al. (2015). They reasoned that iron-rich hematite was the primary light-absorbing mineral in Greenland dust samples, and they simulated three hematite scenarios using low (2.7 %), median (5.6 %) and high (9.3 %) estimates for the proportion of hematite in bulk mineral samples from a literature search on Greenland Ice Sheet surface mineralogy. We refer to these samples as P1, P2 and P3 for low, median and high hematite respectively. The non-hematite component of the sample was a mixture of illite, montmorillonite, calcite, kaolinite, anthropogenic particulates and quartz. Each mineral sample had a lognormal PSD ranging from 0.1 µm to 90 µm. In addition, four "global average" dusts from Flanner et al. (2007), which have typical Saharan optical properties, were included, providing a wide range of dust samples with a range from low to high red-mineral content – critical because these red minerals are most likely to a) have non-negligible albedo-reducing effects, and b) have similarly shaped spectral albedo to glacier algae. We refer to these Flanner et al. (2007) dusts as F1, F2, F3 and F4. We highlight that most of the samples used to generate these bulk optical properties obtained from past literature were from snow, and there may be additional inputs to the bare-ice zone due to ablative release from Holocene age ice. Previous geochemical analyses suggest that the mineral dust on bare-ice surfaces primarily originated from a local source (Wientjes et al. 2011), ruling out any major contribution from windblown Saharan dust. This is corroborated by our mineralogical results, the details of which will be presented in an upcoming paper, and those in past literature indicating that the local bare-ice mineral dust is hematite-poor and rich in felsic mineral phases (Tedesco et al. 2013). While these results and our measured imaginary refractive index indicate a lack of red mineral phases, we included Saharan dusts in our sensitivity study as representative red end-members, thereby accounting for the possibility of occasional inputs of Saharan dusts via atmospheric transport processes.

The ice optical properties in BioSNICAR_GO were also calculated using a parameterisation of geometric optics adapted from van Diedenhoven et al. (2014) instead of the Mie scattering approach taken by the original SNICAR model. A geometrical optics approach to generating ice optical properties was chosen because it enables arbitrarily large ice grains with a hexagonal columnar shape to be simulated, in order to better estimate the albedo of glacier ice where grains are large and aspherical. While the real ice surface is composed of irregularly shaped and sized grains, this approach enabled us to simulate our field spectra much more accurately and circumvented the requirements that individual grains are small and spherical in the case of the Lorenz-Mie approach. The optical properties of the ice grains are modelled using refractive indices from Warren and Brandt (2008). The radiative transfer model is a two-stream model described in full in Cook et al. (2017b) and Flanner et al. (2007). For the radiative transfer modelling presented in this study, the following model parameters were used: Diffuse illumination, ice crystal side-length and diameter per vertical

layer = 3, 4, 5, 8, 10 mm, layer thicknesses = 0.1, 1, 1, 1, 1 cm, underlying surface albedo = 0.15, layer

densities = 300, 400, 500, 600, 700 kg m$^{-3}$. Glacier algae and each of the mineral dusts were added

individually the the upper 0.1 cm layer in mixing ratios of 100, 300 and 500 ppm to quantify their effects on

the surface albedo.

### 2.6 Radiative forcing and biological melt acceleration

The biological radiative forcing was calculated by first differencing the albedo for algal surfaces and the

albedo for clean ice surfaces measured at our field site. This gives the difference in albedo between the clean

and algal ice surfaces, $\alpha_{diff}$. The product of each $\alpha_{diff}$ and the incoming irradiance, $I^*$, provided the

instantaneous power density ($PD_{alg}$) absorbed by the algae. We assume that photosynthetic processes utilise

5% of this absorbed energy – at the upper end of a realistic range for photosynthetic microalgae

(Blankenship et al. 2011; Masojidek et al. 2013). The remainder of $PD_{alg}$ is conducted into the surrounding

ice, giving the instantaneous radiative forcing due to algae ($IRF_{alg}$). Since these cells are coloured by the

purple purpurogallin pigment, we assume the reflective radiative forcing to be negligible, as demonstrated by

Dial et al. (2018). $IRF_{alg}$ was calculated at hourly intervals using incoming irradiance simulated for our field

site using the PVSystems solar irradiance program (https://pvlighthouse.com.au) at 1 nm spectral resolution,

following Dial et al. (2018). The radiative forcing was assumed to be constant between each one hour

timestep, meaning the radiative forcing over one hour ($HRF_{alg}$) could be calculated by multiplying $IRF_{alg}$ by

3600 s h$^{-1}$, assuming that instantaneous radiative forcing is equal to radiative forcing per second. Daily

radiative forcing due to algae ($RF_{alg}$) was then calculated as the sum of $HRF_{alg}$ between 0000 and 2300.

To calculate the algal contribution to melting ($M_{alg}$), $IRF_{alg}$ was multiplied by $10^4$ to convert the radiative

forcing from units of W m$^{-2}$ to W cm$^{-2}$ and then divided by the latent heat of fusion for melting ice (334 J g$^{-1}$)

and integrated over the entire day as described above. This provided a value for the amount of melting

caused by the presence of algae per day assuming the cold content of the ice to be depleted. We calculated

uncertainty by running these calculations for every possible combination of our measured algal and clean ice

spectra and calculating the mean, standard error and standard deviation of the pooled results.

We corroborated these estimates using a point surface energy balance model (Brock and Arnold et al. 2000;

Tedstone, 2019a). This model predicts melting in millimeters of water equivalent given local meteorological

data and information about the ice surface albedo and roughness. We ran this model with the albedo set equal

to the broadband albedo for each CI, $H_{bio}$ and $L_{bio}$ spectrum in our field measurements. The hourly

meteorological data for 21$^{st}$ July 2017 used to force the model was from a Delta-T GP1 automatic weather

station (https://www.delta-t.co.uk/product/ws-gp1/) positioned at our field site. The difference in predicted

melt between the algal surfaces and the clean ice surfaces provided the melt attributed to the presence of

algae. As for the radiative forcing calculations, the uncertainty was calculated by running the energy balance

model for every possible combination of algal and clean ice spectra and calculating the mean, standard error and standard deviation of the pooled results.

**2.7 UAV and Sentinel-2 remote-sensing**

Having quantified algal melt acceleration in localised patches using the methods described in 2.2 – 2.6, we then used a multispectral camera mounted to a UAV to quantify algal coverage across a 200 x 200 m area at our field site. This sample area was kept pristine throughout the study period to minimise artefacts of our presence appearing in the UAV imagery. Inside the sampling area we placed fifteen 10 x 10 cm Ground Control points (GCPs) whose precise location was measured using a Trimble differential GPS. At these

markers we also made ground spectral measurements using an ASD-Field Spec Pro 3 immediately after each flight. The UAV itself was a Steadidrone Mavrik-M quadcopter onto which we integrated a MicaSense Red-Edge multispectral camera. The camera is sensitive in 5 discrete bands with center wavelengths 475, 560, 668, 717 and 840 nm, with bandwidths 20, 20, 10, 10, 40 nm respectively. The horizontal field of view was 47.2° and the focal length 5.4 mm. The camera was remotely triggered through the autopilot which was

programmed along with the flight coordinates in the open-source software Mission Planner (http://ardupilot.org/planner/). Images were acquired at approximately 2 cm ground resolution with 60% overlap and 40% sidelap. The flights were less than 20 minutes long and at an altitude of 30 m above the ice surface.

We applied radiometric calibration and geometric distortion correction procedures to acquired imagery following MicaSense procedures (Micasense 2019; Tedstone, 2019b). We then converted from radiance to reflectance using time-dependent regression between images of the MicaSense Calibrated Reflectance Panel acquired before and after each flight (i.e. a regression line was computed between the reflectance of the white reference panel at the start and end of the flight and used to quantify the change in irradiance during

the flight). Finally, the individual reflectance-corrected images were mosaiced using AgiSoft PhotoScan following procedures developed by the United States Geological Survey (USGS, 2017), yielding a multi-spectral ortho-mosaic with 5 cm ground resolution, georectified to our GCPs. There was generally close agreement between the ground, UAV and satellite-derived albedo although there are some differences that we believe to be the result of different radiometric calibration techniques for satellite, UAV and ground

measurements and the differing degrees of spatial integration will be the subject of an upcoming paper.

To upscale further, we used multispectral data from the Copernicus Sentinel-2 satellite. We selected the 100 x 100 km tile covering our field site (T22WEV) on the closest cloud-free day to our UAV flight on 21[st] July. The L1C product was downloaded from SentinelHub (Sinergise, Slovenia). The L1C product was processed

to L2A using the European Space Agency (ESA) Sen2Cor processor, including atmopsheric correction and reprojection to 20 m resolution.

**2.8 Supervised Classification Algorithms and albedo mapping**

To map and quantify spatial coverage of algae over the ice-sheet surface we employed a supervised classification scheme. A RF classifier was trained on the field spectra collected on the ice surface (see section 2.3) and then applied to multispectral images gathered by the UAV and Sentinel-2. We also included spectra obtained at the same field site in July 2016 to our training set, giving a total of 231 labelled spectra. A schematic of the classification workflow is provided in Supp Info 3. Our HCRF measurements were first

reduced to reflectance values at five key wavelengths coincident with the centre wavelengths measured by the MicaSense Red-Edge camera mounted to the UAV (blue: 0.475, green: 0.560, red: 0.668, red-edge: 0.717, NIR: 0.840 μm) yielding reflectance at each wavelength as a feature vector for the classifier. The classification labels were the surface type as determined by visual inspection: SN (snow), CI (clean ice), CC (cryoconite), WAT (water), $L_{bio}$ (low biomass algae) and $H_{bio}$ (high biomass algae). For the algal surface

classes our visual assessment was corroborated by microscopy as described in section 2.2. This data set was then shuffled and split into a training set (70%) and a test set (30%). The training set was used to train three individual supervised classification algorithms: Naive Bayes, K-nearest neighbours (KNN) and support vector machine (SVM). For the SVM, the parameters C and gamma were tuned using grid search cross validation. Two ensemble classifiers were also trained: a voting classifier that combined the predictions of

each of the three individual classifiers, and a RF algorithm. The performance of each classifier was measured using precision, accuracy, recall and F1 score and also by plotting the confusion matrix and normalised confusion matrix for each classifier. In all cases the RF outperformed the other classifiers according to all available metrics (Supp Info 4). The performance of the RF classifier was finally measured on the test set, demonstrating the algorithm's ability to generalise to unseen data outside of the training set. Overfitting is

not usually associated with the RF classifier, and the strong performance on both our training and test sets confirms that the model generalizes well. For these reasons, we used the RF algorithm to classify our multispectral UAV and Sentinel-2 images. Training the classifier using data from field spectroscopy ensures the quality of each labelled datapoint in the training set, since our sampling areas were homogenous and surface samples analysed in the laboratory, circumventing issues of spatial heterogeneity and uncertainty in

labelling that could lead to ambiguity for direct labelling of aerial images. Simultaneously to the surface classification, we calculated the albedo in each UAV pixel using the narrowband to broadband coversion of Knap et al. (1999) applied to the reflectance at each of the five bands.

This protocol was repeated for Sentinel-2 imagery. Additional bands are available for use as feature vectors

in the case of Sentinel-2. Directional reflectance data was reduced to eight bands coincident with the centre wavelengths measured by Sentinel-2 at 20m ground resolution (0.480, 0.560, 0.665, 0.705, 0.740, 0.788, 0.865, 1.610 μm). Training on reduced hyperspectral data is a novel approach which has several advantages over gathering multispectral data: first, the method is sensor-agnostic because the classifier can be retrained

with a diffent selection of wavelengths for other upscaling platforms, enhancing the reuseability of the field measurements; second, we have confidence in our labels because each sample has been laboratory analysed to confirm its composition, reducing label ambiguity; finally, the limited field of view of the field spectrometer reduces error arising from mixing of spectra from heterogeneous ice surfaces. Sentinel-2 imagery was masked using the MeASUREs Greenland Ice Mapping Project ice mask (https://nsidc.org/data/nsidc-0714) to eliminate non-ice areas. Pixels with more than 50% probability of being obscured by cloud were masked using the Sentinel-2 L2A cloud product generated by the Sen2Cor processor. For the calculation of albedo in each pixel, the additional bands available in the Sentinel-2 images enabled the application of Liang et al.'s (2001) narrowband to broadband conversion.

**2.9 Comparing 2016 and 2017**

In 2017, the GrIS Dark Zone had relatively small spatial extent, high albedo and short duration in comparison to the other years in the MODIS record, particularly since 2007, whereas the Dark Zone was especially dark, widespread and prolonged in 2016 (Fig 6; Tedstone at al. 2017). We therefore conducted a comparison between the algal coverage on the same dates in 2016 and 2017. First, we examined variations in the extent and duration of the Dark Zone along with snow depths and snow clearing dates for the south western ablation zone using MODIS, extending the time series of Tedstone et al. (2017). Bare ice was mapped by applying a threshold reflectance value (R < 0.60 at 0.841-0.871 µm) to the MOD09GA Daily Land Surface Reflectance Collecton 6 product. Within the bare-ice area, dark ice was mapped using a lower reflectance threshold (R < 0.45 at 0.62 – 0.67 µm). The area of interest was the "common area" defined by Tedstone et al. (2017) bounded within the latitudinal range 65 – 70º N, and is equal to that used by Wang et al. (2018). To measure the annual dark ice extent (in km$^2$) we counted the pixels that were dark for at least 5 days each year. The annual duration was defined at each pixel as the percentage of daily cloud-free observations made in each JJA period which were classified as dark. The timing of bare ice appearance was calculated from MODIS using a rolling window approach on each pixel (see Tedstone et al. 2017). The mean snow depths were extracted from outputs from the regional climate model MAR v3.8 (Fettweis et al. 2016) run at 7.5 km resolution forced by ECMWF ERA-Interim reanalysis data (Dee et al. 2011). These data enabled a comparison of the extent and timing of dark ice in 2016 and 2017.

To examine algal coverage in each year we identified the Sentinel-2 tile covering our field site (22WEV) on the closest cloud-free date to the UAV flight day (21$^{st}$ July) in each year. These were 28$^{th}$ July 2017 and 25$^{th}$ July 2016. Since we are interested in the bare-ice zone, snow covered pixels were omitted from the calculations.

**2.10 Runoff Modelling**

Runoff at the regional scale was calculated using van As et al.'s (2017) SMB model forced with local automatic weather station and MODIS albedo observations (van As et al., 2012; 2017). The model interpolates meteorological and radiative measurements from three PROMICE automatic weather stations on the K-Transect (KAN_L, KAN_M and KAN_U) and bins them into 100 m elevation bands (0 to 2,000 m a.s.l.). Surface albedo is from MODIS Terra MOD10A1 albedo and is averaged into the same 100 m elevation bins. For every one-hour time step, the model iteratively solves the surface energy balance for the surface temperature. If energy components cannot be balanced due to the 0 C surface temperature limit, a surplus energy sink for melting of snow or ice is included. If surface temperature is greater than the melting point, the surplus energy is used for melting of snow or ice. When calculating turbulent heat fluxes, aerodynamic surface roughness for momentum was set to 0.02 and 1 mm for snow and ice, respectively (after van As et al. 2005; 2012; Smeets and Van den Broeke, 2008). We extrapolate modelled runoff across the south-western GrIS (65 – 70 ° N) by deriving the areas of each elevation bin using the Greenland Ice Mapping Project (GIMP) DEM (Howat et al., 2014). Total summer runoff from bare ice was calculated by summing runoff in elevation bins that had mean daily albedo of less than 0.60. Total summer runoff from dark ice only was calculated in the same way but using a 0.39 threshold. In van As et al.'s (2017) study they compared the performance of the model with independent observations and found errors to be negligible in the bare-ice zone.

To determine the algal contribution to runoff, we used Equation 1:

$$R_{alg} = R_{tot} * ((M_{Hbio} * C_{Hbio}) + (M_{Lbio} * C_{Lbio})) \qquad (Eq.1)$$

where $R_{alg}$ = Runoff due to algae, $R_{tot}$ = the total runoff from the bare-ice zone calculated using our runoff model, $M_{Hbio}$ and $M_{Lbio}$ = mean percentage of total melt attributed to algae in $H_{bio}$ and $L_{bio}$ areas as calculated by our energy balance modelling described in 2.6, $C_{Hbio}$ and $C_{Lbio}$= the proportion of $C_{tot}$ comprised of $H_{bio}$ and $L_{bio}$ areas in our UAV or Sentinel-2 images. As discussed later, the Sentinel-2 algal coverage estimate is conservative because it often fails to resolve $H_{bio}$ surfaces and therefore provides a lower bound on the runoff attributed to algae. An upper bound was therefore also calculated by assuming the spatial coverage derived from our UAV remote-sensing – which can accurately distinguish $L_{bio}$ and $H_{bio}$ - surfaces is representative of the south-western Dark Zone. We were thereby able to estimate upper and lower limits for the runoff attributed to algal growth on the south western ablation zone.

## 3. Results and Discussion

### 3.1 Algae reduce ice albedo

The ice surfaces we studied were divided into four classes depending upon the algal abundance measured in the melted ice samples: High algal abundance ($H_{bio}$), Low algal abundance ($L_{bio}$), Clean Ice (CI) and Snow (SN). The algal abundance in each class was as follows: $H_{bio}$ = 2.9 x $10^4$ ± 2.01 x $10^4$, $L_{bio}$ = 4.73 x $10^3$ ± 2.57 x $10^3$, CI = 625 ± 381, SN = 0 ± 0 (1 SD). These cell abundances were significantly different between the classes (one-way ANOVA, F = 10.21, p = 3 x $10^{-5}$) which Bonferroni-corrected t-tests indicated to be due to variance between all four groups. The dominant species of algae were *Mesotaenium bergrennii* and *Ancylonema nordenskioldii* (Fig 2B), confirming observations made by Stibal et al. (2017) and Williamson et al. (2018) in the same region. Their long, thin and approximately cylindrical morphology has been shown to be near-optimal for light absorption (Kirk, 1976). The albedo of the ice surface also varied significantly between the surface classes (one-way ANOVA for broadband albedo: F= 7.9, p = 2.8 x $10^{-4}$), again with Bonferroni-corrected t-tests showing variance between all four groups (Supp Info 5 C,D). Greater algal abundance was associated with lower albedo, with the albedo reduction concentrated in the visible wavelengths (Fig 2C) where both solar energy receipt and algal absorption peak (Cook et al., 2017b; Williamson et al., 2018), diminishing towards longer near infra-red (NIR: > 0.70 µm) wavelengths where ice absorption, represented by the effective grain size, is most likely to cause albedo differences (Warren, 1982). A strong inverse correlation (Pearson's R = 0.75, p = 2.74 x $10^{-9}$) was observed between the natural logarithm of algal cell abundance (cells $mL^{-1}$) in the surface ice samples and broadband albedo (Fig 2D). The linear regression coefficient of determination between the albedo and the natural logarithm of cell abundance was 0.57. It is unsurprising that the cell abundance does not account for all variation in albedo because there are also albedo-reducing effects related to the physical structure of the ice and presence of melt water (as demonstrated for snow by, for example, Warren, 1982). An inverse relationship was also observed between broadband albedo and biovolume (calculated as the sum of the products of the mean measured cell volumes and the cell counts for each algal species) but the coefficient of determination was lower ($r^2$ = 0.42). This may well be the result of larger cells having a smaller effect on albedo than more numerous, smaller cells for a given total volume. The relationship between absorption and scattering coefficients and cell size may also not be straightforward for algal cells due to an increasingly important contribution to the cell optical properties from internal heterogeneity, organelles, cell walls and the pigment packaging effect in larger cells (Morel and Bricaud, 1981; Haardt and Maske, 1987).

The albedo of $H_{bio}$ and $L_{bio}$ surfaces is depressed in the visible wavelengths (0.40 - 0.70 µm, Fig 2B), creating a 'red-edge' spectrum commonly used in other environments as a marker for photosynthetic pigments (Seager et al., 2005) and for mapping algae over the GrIS by Wang et al. (2018). Chlorophyll-a has a specific absorption feature at 0.68 µm which is hard to discern in the raw spectra, but clear in the derivative spectra (Fig 3A) for $H_{bio}$ and $L_{bio}$ but not CI and SN. This feature has previously been described as "uniquely biological" (Painter et al., 2001) and supports the hypothesis that the albedo reduction observed in these samples is primarily due to algae. Our measurements therefore strongly indicate a biological role in reducing the albedo of the GrIS surface; however to test that the lower broadband and spectral albedo observed on

algal surfaces is primarily due to the presence of algal cells, it was also necessary to compare the albedo-reducing effects of the algae to that of local mineral dust.

**3.2 Algae have greater impact on albedo than mineral dust**

We used radiative transfer modelling to compare the albedo-reducing effects of local mineral particles and algal cells (Fig 3B). To compare algal and mineral albedo-reducing effects, the model was run with fixed irradiance and ice physical properties that were chosen to reduce the absolute error between the simulated albedo for ice without any impurities and our mean measured clean ice spectrum and the optical properties of local mineral dust and algal cells were incorporated into the model individually. The imaginary refractive index of the mineral dust sample (Fig 3C) was lower than Skiles et al's (2017) dust samples and the other dust samples included in our sensitivity study, indicating a greater prevalence of weakly absorbing minerals and scarcity of red minerals in the bare ice. The mass absorption coefficient, asymmetry parameter and single scattering albedo (Fig 4 A, B, C) were broadly comparable to those of Skiles et al.'s (2017) finest dust sample which had a log-normal PSD with diameters ranging between 0.1 and 1µm (that sample was identical to the dust sample F1 included in this study). The majority of the mineral mass in the field sample was composed of very small particles (Fig 4D), and an optically thick layer of the minerals had a near-white colour to the naked eye. While the mean diameter was 1.86 µm, this was influenced by the presence of a few very large fragments. 75% of the particles had diameter < 1.68 µm, and 50% had diameter < 0.56 µm. The effect of adding these local mineral dust to a simulated ice column varied depending upon the dust sample used (Fig 3B). For example, adding a hypothetical 300 µg/g of mineral dust to the upper 1 mm of ice caused a small increase in albedo (0.008) relative to clean ice. This was also the case for dusts F1 and F2, which caused an albedo increase of 0.01 and 0.002 respectively. The same mass mixing ratio of the P1, P2 and P3 dusts caused a small albedo reduction: 0.0005 (P1), 0.003 (P2) and 0.01 (P3). In contrast, adding 300 µg/g of algae decreased the albedo by 0.02. We present the albedo change resulting from 100, 300 and 500 ug/g mixing ratios of all the dust species and glacier algae in our sensitivity study in Table 1. There was no scenario where the addition of glacier algae could increase the albedo of the ice, and their albedo reducing effect always increased with additional biomass. The albedo reduction due to hematite-rich mineral dusts is concentrated into the shorter VIS wavelengths, as it is for algae. However, the spectral shape of ice with dust differs from ice with algae. For hematite-rich dust, the ice albedo increases with wavelength up to 0.70 µm; for algae, the spectrum is much flatter or even downsloping with increasing wavelength between 0.35 and 0.45 µm before rising steeply to the "chlorophyll-a bump" at 0.55 µm, and a gentle increase to 0.70 µm. These spectral features are consistent with our field spectra for algal ice, and can only be recreated in the radiative transfer model when algal particles make up the majority of the surface impurity load.

These radiative transfer simulations indicate that mineral dust is unlikely to be directly causing the albedo decline on the GrIS, although they may influence the ice albedo indirectly by acting as substrates for the

formation of low-albedo microbial-mineral aggregates known as cryoconite granules, which are often found in quasi-cylindrical melt holes or scattered over ice surfaces (Wharton et al. 1985; Cook et al. 2015a) or by providing a nutrient source stimulating algal growth (Stibal et al. 2017). This is especially true because there is evidence in the previous literature that the dust present on the GrIS bare-ice surface are likely derived from a local source with no contribution from Asian dusts or volcanic ash (Wientjes et al. 2011) and that red minerals such as hematite, geothite and ilminite are present only in very low concentrations (Wientjes et al. 2011; Tedesco et al. 2013; Sanna and Romeo, 2018) that would have a negligible effect on the ice optics. This was also supported by our preliminary mineralogical analysis of our local dust samples, implying that the minerals included in our sensitivity test are truly extreme upper end-members for albedo reduction and 'redness' caused by dust deposition in this region, and that mineral dusts are not responsible for albedo decline on the south-western GrIS.

The minimal direct albedo reducing effect from local minerals on the ice surface is seemingly in contrast to some previous studies such as Wientjes et al. (2010; 2011) and Bøggild (2010); however, we highlight that neither of the Wientjes et al. (2010; 2011) studies directly measured the surface albedo or any optical properties of the mineral dusts retrieved from their GrIS sampling sites and only inferred mineralogical darkening from low spectral resolution MODIS data and the presence of a "wavy pattern" observed across the dark zone. We argue that while the "wavy pattern" may be indicative of geological outcropping onto the ablation zone, it does not necessarily follow that these minerals are responsible for surface darkening. In support of this, Wientjes et al. (2011) found strongly scattering and weakly absorbing quartz to be the dominant mineral in surface ice and speculated that biota may be having a darkening effect. Bøggild et al. (2010) found mineral dust to be an albedo reducer in Kronprinz Christian's Land (80N, 24W) but this area is geologically and climatologically distinct from our field site, and their transect only spanned ~8 km from the ice-sheet margin, being an area prone to local dust deposition. In this study, we have demonstrated using empirical measurements and radiative transfer modelling that algae are potent albedo reducers and mineral dusts are not, at least in this region where small, strongly scattering and weakly absorbing mineral fragments sit atop relatively dark bare ice. These findings are consistent with several previous studies (Stibal et al. 2017; Yallop et al. 2012) that found mineral dust to be insignificant for explaining albedo variations in the same region or that algal cells had a greater albedo reducing effect than mineral dusts in north-west Greenland (Aoki et al. 2013).

**3.3 Indirect effects of algae**

Algae predominantly reduce the ice albedo in the visible wavelengths (0.40 – 0.70 µm), whereas variations in the NIR result mainly from changes to ice grain radii and the presence of liquid water (Warren, 1982; Green et al., 2002). We compared the area of an absorption feature (i.e. the sum of the distances between a straight line drawn between the shoulders – 0.95 µm and 1.035 µm - of the absorption feature and the albedo

at each wavelength) centered at 1.02 μm between the different surface types, finding significant differences between all four surface classes (one-way ANOVA, F = 12.8, p =7.16 x 10[-7]) driven predominantly by variations between the two algal surfaces and the two clean surfaces. This absorption feature is linked to the optical properties of snow because it scales with grain size (Nolin and Dozier, 2000), so we interpret these variations as evidence that the optical properties of the ice surface differed between the surface classes, having an effect on the measured albedo. The feature area is smallest for $H_{bio}$ followed by $L_{bio}$, CI and largest for SN (Supp Info 5B). The features with the smaller areas also had lower albedo minima. The absorption features are also subtly, but systematically, left-asymmetric for the algal surfaces, consistent with the presence of liquid water in the fast-melting ice beneath algal blooms (Green et al., 1998; Cook et al., 2017b).

These observations, along with the linear regression coefficient of determination of 0.57 between albedo and the logarithm of cell abundance suggest that the lower albedo of algal surfaces is not explained entirely by enhanced absorption due to algae, but also by the smoother, wetter ice surface with fewer opportunities for high-angle scattering of photons (Jonsell et al. 2003), compared to the well-drained and porous CI surfaces. The spatial and temporal development of the weathering crust is therefore a major control on bare-ice albedo (Muller and Keeler, 1969; Jonsell et al. 2003). Cause and effect is unclear because algae may cause this by enhancing melting of the weathered surface or may grow preferentially where there is already more melt. We expect the explanation to be a combination of these two interlinked processes, especially since melting of ice liberates nutrients that stimulate algal growth. Furthermore, our radiative transfer model indicates that enhanced absorption of solar radiation at the ice surface due to glacier algae reduces the energy penetrating into the ice, reducing internal erosion, while surface lowering is enhanced due to emission of absorbed solar energy as heat, promoting thinner, less porous weathering crusts with more interstitial water and feeding back to lower albedo underlying ice. We therefore highlight the role of indirect feedbacks (Cook et al., 2017a,b; Tuzet et al., 2017) in biological darkening of ice-sheets. This process is self-amplifying because algal growth is stimulated by melt, which can be enhanced by algal growth (Yallop et al., 2012; Ganey et al., 2017; Stibal et al., 2017; Cook et al., 2017a,b; Dial et al., 2018), enhancing the albedo lowering process - an example of a biocryomorphic process where biota alter the physical, chemical and hydrological conditions of the ice surface with beneficial consequences to the biota (Cook et al. 2015b).

### 3.4 Algae enhance radiative forcing and melt

Having determined that glacier algae reduce the ice surface albedo, we took an empirical approach to quantifying their impact upon energy balance following Ganey et al. (2017), which includes both direct albedo effects (enhanced absorption of shortwave solar radiation by the algal cells) and the indirect effects explained above. Integrated over the entire day, this indicated a daily mean biological radiative forcing of 116 W m[-2] and 65 W m[-2] for $H_{bio}$ and $L_{bio}$ surfaces respectively, similar to RFs for Alaskan snow algae calculated by Ganey et al (2017). We used the biological radiative forcing integrated over the entire day

(Supp Info 5A) and the latent heat of fusion for ice (334 J cm$^{-3}$) to estimate $1.35 \pm 0.01$ (S.E) cm w.e. of

melting due to algae in $H_{bio}$ areas on 21$^{st}$ July. For $L_{bio}$ sites, biological melting on 21$^{st}$ July 2017 was $1.01 \pm$

0.01 (S.E) cm w.e.

We corroborated this estimate using a point surface energy balance model (Brock and Arnold et al. 2000).

The melt attributed to the presence of algae predicted by the energy balance modelling method was similar to

that predicted using the radiative forcing method, with $1.37 \pm 0.48$ (S.E) cm w.e. attributed to $H_{bio}$ and $0.95 \pm$

0.41 (S.E) cm w.e. attributed to $L_{bio}$. Expressing the melt attributed to algae as a proportion of the total

melting in the algal sites gives $26.15 \pm 3.77$ % (S.E) of the local melting attributed to algae in the $H_{bio}$

surfaces and $21.62 \pm 5.07$ % (S.E) for $L_{bio}$ surfaces.


**3.5 Algae are widespread across the south-western ablation zone**

Our analyses demonstrate that algae have a dramatic darkening effect on the ice surface, leading to increased

melting. However, the importance of this effect depends upon the spatial extent of the algal blooms over

thousands of kilometers. To determine spatial coverage at our field site we aclassified multispectral images

acquired from a UAV flown over a 200 x 200 m area. The classified UAV image indicated that 78.5% of the

area was covered by algal blooms of which 61.1% was $L_{bio}$ and 17.4% was $H_{bio}$ (Table 3; Fig 5). The high

ground resolution of the imagery enabled a qualitative assessment of the algorithm performance by visual

comparison between the classifier and the raw imagery (following Ryan et al. 2018a). The algorithm

produced qualitatively realistic bloom shapes, correctly placed water in channels and individual cryoconite

holes in their correct positions. The confusion matrix indicates that occasional misclassifications are

generally between water and cryoconite (Supp Info 6). This is unsurprising since both cryoconite and water

have relatively flat spectral shapes with few spectral features and cryoconite is often found beneath pools of

surface water. We also point out that our cryoconite spectral reflectance measurements were made with

cryoconite filling the entire field of view of the spectrometer, so best represent large cryoconite holes or

dispersed cryoconite rather than surfaces peppered with many small holes. There was also some ambiguity

between thin, wet snow and bare glacier ice, as these surfaces are spectrally similar. Nevertheless, these

misclassifications affect a small area of the pixel and do not affect our estimate of algal bloom coverage.

We also classified Sentinel-2 satellite data (Fig 5). The confusion matrices (Supp Info 6) indicate similar

misclassification types and frequencies to the UAV model. The predicted algal coverage was 58.87%. $H_{bio}$

surfaces were much less common than $L_{bio}$ ($H_{bio} = 2.53$ % , $L_{bio} = 56.54$%, Table 3). The spatial coverage by

algae was different in the Sentinel and UAV datasets especially for $H_{bio}$, likely because a) the Sentinel-2

imagery includes ice that is outside of the Dark Zone, raising the overall reflectivity, and b) even in the UAV

image, which was retrieved from within the Dark Zone, $H_{bio}$ surfaces comprise just 17% of the ice surface

and have a patchy distribution. The lowest albedo surfaces – cryoconite and water – cover a small fraction (<

3%) of the total area in both UAV and Sentinel-2 images (Table 3), although we note that many individual cryoconite holes will not be detected as they are smaller than the spatial resolution of either Sentinel-2 or the UAV. The spatial coverage reported here from our multispectral UAV imagery is consistent with a k-nearest

neighbours classification scheme applied to RGB (Red, Green, Blue) imagery from a fixed wing UAV flight over the Kangerlussuaq region by Ryan et al. (2018a). They found up to 85% of the ice surface to be composed of 'ice containing uniformly distributed impurities' in the same region of the Dark Zone in July 2014, which our observations confirm were dominated by algae. They also found < 2% of the ice surface to be cryoconite covered and water coverage was < 5% (except for a supraglacial lake in their imaged area).

This analysis demonstrates that algae are a major component of the ice surface. The larger spatial coverage of algae observed in UAV images compared to Sentinel-2 images likely results from spatial integration occurring at the coarser spatial resolution associated with Sentinel-2 data, where pixels are likely to be classified as CI unless the majority of the pixel is algae-covered. Smaller $H_{bio}$ patches are rarely detected presumably because they are unlikely to cover the majority of a 20 m pixel. The higher detection limit for

algae with decreasing ground resolution makes our estimate of spatial coverage from Sentinel-2 conservative. We highlight that this will have a much larger effect on studies aiming to quantify cell abundance using Sentinel-3 where the ground resolution is 300 m.

### 3.6 Algae reduce the ice albedo across the south-western ablation zone


There was a significant difference between the albedos of each surface class in all four datasets, consistent with the findings from our ground spectroscopy (Fig 5; Table 2). The albedo of each surface class is approximately consistent between the datasets, despite the variation in spatial coverage, giving confidence in the accuracy of our remote-sensing albedo retrievals and the classification algorithm. In the expansive areas

where algae are present (Fig 5) the ice albedo is on average 0.13 lower for $L_{bio}$ and 0.25 lower for $H_{bio}$ compared to clean ice (Table 2). This, combined with our ground-based spectroscopy, radiative forcing calculations, radiative transfer and energy balance modelling, provides robust evidence in support of algae having a significant melt-accelerating effect on the GrIS. We cannot yet explicitly separate mineral and biological effects, but our theoretical and empirical analyses indicate that: a) local mineral dust cannot

explain the observed albedo reduction, b) low-albedo areas had significantly elevated algal cell numbers relative to clean ice, c) uniquely biological features were detectable in the spectra and derivative spectra for the lower albedo sites, and d) radiative transfer models incorporating algal cells with realistic pigment profiles demonstrate the mechanism of albedo reduction. These observations confirm that supervised classification of $H_{bio}$ and $L_{bio}$ surfaces is indeed detecting surfaces with high algal loading and can be used to

estimate algal bloom extent. Again, we point out that this estimate is conservative because there is certain to be glacier algae present in low numbers in some of the areas that are classified as clean, and $H_{bio}$ patches are not detected where the area of the patch is much smaller than the ground resolution of the sensor.

Furthermore, these calculations consider the total albedo-reducing effect, inclusive of ice structure and meltwater feedbacks, not only the direct light-absorbing effects of the algal biomass.

**3.7 Algae cause enhanced GrIS runoff**

We ran a surface mass balance (SMB) model forced with local automatic weather station and MODIS albedo observations (van As et al., 2012) to estimate 45.5 Gt runoff from all bare ice and 33.8 Gt from dark ice in 2017. We used the mean spatial coverage determined using our remote-sensing in each year and our radiative forcing calculations that attributed $21.62 \pm 5.07$ (standard error) % of melting to algae in $L_{bio}$ sites and $26.15 \pm 3.77$ (standard error) % in $H_{bio}$ sites to generate estimates for the GrIS runoff caused by algal growth. We have provided upper and lower estimates based on our two remote-sensing datasets, because while our UAV is able to accurately map $H_{bio}$ and $L_{bio}$ surfaces, we cannot be certain that the spatial coverage derived from the 200 x 200 m area is representative of the south-western Dark Zone. At the same time, our Sentinel-2 remote-sensing underestimates algal coverage because $H_{bio}$ patches are often too small to be resolved at 20 m pixel resolution. Therefore, we used the spatial coverage determined by our Sentinel-2 classification as a lower bound, and spatial coverage determined by our UAV classification as an upper bound on our estimate of total runoff attributed to the presence of algae.

We found that in 2017 between 4.4 – 6.0 Gt of ice loss could be attributed to the growth of algae, representing 10 - 13 % of the total runoff from the south-western GrIS, with the lower estimate generated using algal coverage from Sentinel-2 and the upper estimate generated using spatial coverage at our field site from our UAV. When the the calculations to runoff were restricted to the Dark Zone only (i.e. excluding areas in the ablation zone not classified as "dark") algal contributions to total runoff were  up to 18 %. These calculations confirm that algal growth is an important factor in the contribution of the GrIS to global sea level rise. This contribution will increase if biologically-darkened areas expand or a greater proportion of the ice is covered by high biomass blooms under warmer climates. These observations therefore indicate that the omission of biological growth is leading current models to underestimate future GrIS contributions to sea level rise.

**3.8 Interannual variability and potential positive feedback**

MODIS data (Fig 6) indicates that 2017 was a particularly high albedo year when the Dark Zone was especially small and bright, whereas 2016 was a particularly low-albedo year where the dark zone was wider and darker than most years (Fig 6 A,B and Tedstone et al. 2017). Previous field evidence (Williamson et al., 2018) demonstrates that the ice was darkened by high concentrations of algae in 2016. In our Sentinel-2 remote-sensing tile (22WEV) the bare-ice zone was wider in 2016 (6758 km$^2$) than in 2017 (6205 km$^2$), and a larger area was covered with algae  (on 25$^{th}$ July 2016, 3919 km$^2$ was covered by algae compared to 3653 km$^2$ on 28$^{th}$ July 2017). While the proportional total algal coverage was similar between the two years (57.99 % in 2016, 58.87 % in 2017), the proportion of the algal ice that was classified $H_{bio}$ was much higher in 2016

(8.35 %) compared to 2017 (2.54 %). The mean albedos and their standard deviations were very similar for each ice surface class in both years (Table 2). The runoff from the south-western GrIS bare ice (albedo < 0.6) was 94.1 Gt in 2016, of which 67.6 was from dark ice (albedo < 0.39). We estimate that 8.7 – 12.2 Gt of this runoff was attributable to the growth of algae, representing 9 – 13 % of the total runoff from bare-ice sectors. The absolute values for runoff are therefore much higher but the proportion of the bare-ice total attributed to algae was approximately the same between the two years.

The snow line retreated further, earlier in 2016 compared to 2017, creating a wider bare-ice zone that existed for longer and was not transiently covered by summer snowfall events, whereas in 2017 a smaller bare-ice area was exposed later and was covered by 5 - 10 cm of snow several times during the summer (Fig 6 C, D). The more prolonged exposure of a larger bare-ice zone in 2016 enabled $L_{bio}$ surfaces to extend to higher elevations and biomass to accumulate to greater mass concentrations at lower elevations in summer 2016, explaining the greater $H_{bio}$ coverage. This indicates that the intensity of the algal bloom is a function of exposure time, as postulated by Tedstone et al. (2017) and Williamson et al. (2018). More prolonged exposure of larger ablation areas under a warming climate (Stroeve et al. 2013; Shimada et al. 2016; Tedesco et al. 2016; Tedstone et al. 2017) are likely to be prone to more spatially expansive, darker algal blooms that enhance melt rates, leading to a potential positive feedback that is not currently accounted for in surface mass balance models whereby earlier exposure of bare ice leads to enhanced algal coverage, which will be able to accumulate higher biomass, and accelerate melting. Melting, in turn stimulates algal growth by liberating nutrients and liquid water.

**4. Conclusions**

Our measurements and modelling demonstrate that the growth of algae on the GrIS is accelerating the rate of melting and increasing the GrIS contribution to global sea level rise. Our field spectra show a dramatic depression of the surface albedo in the visible wavelengths for surfaces contaminated by algae. Derivative analysis of the same spectra show uniquely biological absorption features and an inverse relationship was observed between biomass and surface albedo. We employ a novel radiative transfer model to show that this albedo decline cannot be attributed to local mineral dusts. Radiative forcing calculations and an energy balance model predict that melting of glacier ice can be accelerated by 21.62 ± 5.07 (SE) % for $L_{bio}$ surfaces and 26.15 ± 3.77 (SE) % for $H_{bio}$ surfaces. We demonstrate that the growth of algae occurs over a large proportion of the ablating area of the south western GrIS by identifying algal blooms in remote-sensing data from a UAV and Sentinel 2, finding 78.5 % of the surface within a  200 x 200 m sample area at our field site to be algae covered. Using Sentinel-2 we detected algae covering 57.99 % of the Kangerlussuaq region in 2017 and 58.87 % of the same region in 2016. The spatial resolution of the sensor makes these conservative estimates, especially for $H_{bio}$ surfaces. Runoff modelling informed by our field measurements and remote-sensing estimate between 4.4 and 6.0 Gt of runoff from the south western ablation zone could be attributed to the growth of algae in summer 2017, representing 10- 13% of the total. Because 2017 was a particularly high

albedo year for the south western GrIS, we also ran our analysis for the particularly low-albedo 2016 melt season. In 2016 a wider bare-ice zone was exposed for longer, and there was a concomitant increase in the extent of the algal bloom, more of which was classified as $H_{bio}$ (high biomass). The percentage algal

contribution to south western GrIS runoff was approximately the same as in 2017 (9 – 13 %) but the absolute volume was much higher (7.3 – 11.9 Gt). This interannual comparison indicates the existence of a feedback because in years where snow retreats further, earlier, there is a larger and more prolonged area for algal bloom development where melting is enhanced, stimulating further algal growth. This study therefore demonstrates that algae are important albedo-reducers and cause a melt-enhancing feedback across the south-

western GrIS. The omission of these critical biological albedo feedbacks from predictive models of GrIS runoff is leading to underestimation of future ice mass loss and contribution to global sea level rise. This is particularly significant because larger ablation zones and longer growth seasons are expected in a future warmer climate.

**5 Data Availability**

Codes and datasets used in this study are available at the following doi's:
BioSNICAR_GO code and data: [www.github.com/jmcook1186/BioSNICAR_GO](www.github.com/jmcook1186/BioSNICAR_GO)
Ice Surface Classification codes: [www.github.com/IceSurfClassifiers](www.github.com/IceSurfClassifiers)

Spectra Processing codes: [www.github.com/jmcook1186/SpectraProcessing](www.github.com/jmcook1186/SpectraProcessing)
Field and associated data: [www.github.com/jmcook1186/Data_Archive_TC2019](www.github.com/jmcook1186/Data_Archive_TC2019)

**6 Author Contribution Statement**

JC developed the measurement protocol, gathered field measurements, analysed the data, wrote the main codes, curated the data repository, produced the figures and wrote the manuscript. OMcA was instrumental in building and testing the UAV. AT, CW, JMcC, SH gathered field data. CW provided advice regarding microscopy and biological sampling protocols and helped with experimental design, and also led the empirical measurements of glacier algae pigmentation and absorption coefficients. AT wrote the code for

radiometric calibration of multispectral imagery from the UAV and post-processed the UAV images, derived 2016 and 2017 dark ice extent from MODIS imagery, analysed MAR snow depth outputs and produced Fig 6, translated the energy balance model into Python and made significant contributions to the manuscript writing and experimental design. JMcC and SM provided cleaned mineral dust and PSD data to feed into the radiative transfer model and JMcC provided useful discussions regarding experimental design. MS provided

DISORT modelling and estimates of mineral dust refractive indices. MF helped develop the bio-optical model. RB provided advice regarding field spectroscopy and helped measure mineral dust refractive indices in the laboratory. AJH helped develop the experimental design. AH, JR and AMcG both provided advice on UAV remote-sensing. JR, DvA and AH modelled runoff from the GrIS dark zone. AD provided microscopy

images from field samples. Other authors contributed to field work and/or sample preparation and
commented on the style and content of the final manuscript.

## 7 Acknowledgements

JC, AT, AJH, CW, AD, SH, AmcG, AA, TDLIF, EH, MY and MT acknowledge funding from UK National Environmental Research Council Large Grant NE/M021025/1 'Black and Bloom'. JC gratefully acknowledges the Rolex Awards for Enterprise, National Geographic and Microsoft ("AI for Earth"). LGB, JMcC and JBM acknowledge funding from the UK National Environmental Research Council Large Grant NE/M020770/1 'Black and Bloom' and LGB and SM acknowledge funding from the German Helmholtz Recruiting Initiative (award number I-044-16-01). TG acknowledges the Gino Watkins Memorial Fund and Nottingham Education Trust. Greenland Analogue Project (GAP) weather station data are made available through the Programme for Monitoring of the Greenland Ice Sheet ([www.promice.dk](http://www.promice.dk)). MAR v3.8.1 regional climate model outputs used estimate mean snow depth were provided by Xavier Fettweis.

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

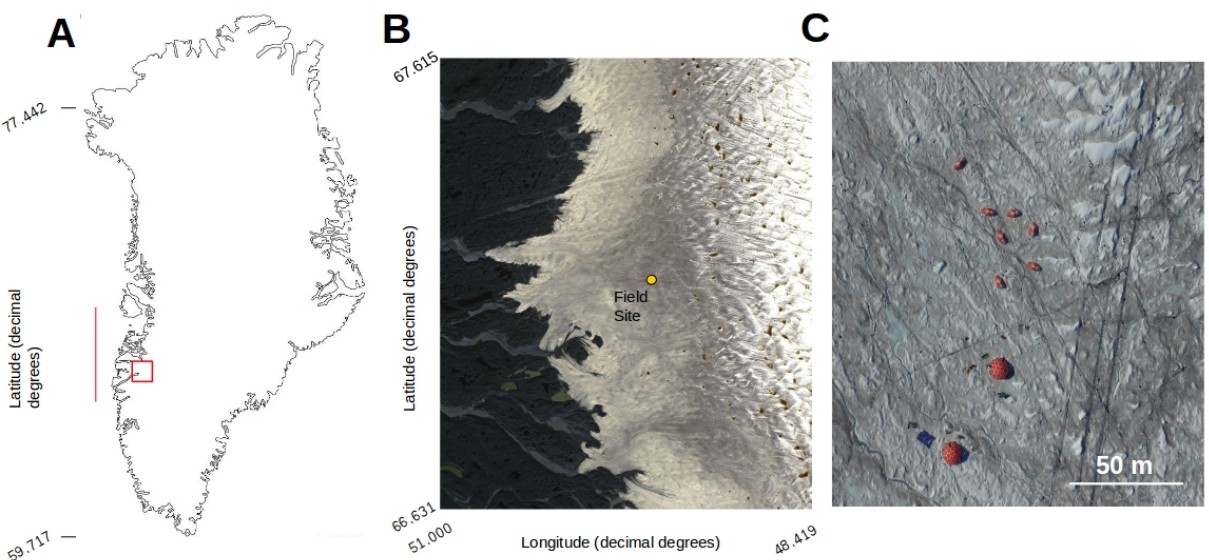


Fig 1: A) Map of Greenland showing the bounding box of the Sentinel-2 tile containing our field site (red box) and the latitudinal extent of our runoff modelling (red line). The area in the red box is presented in detail in B) with our field site marked with a yellow dot.


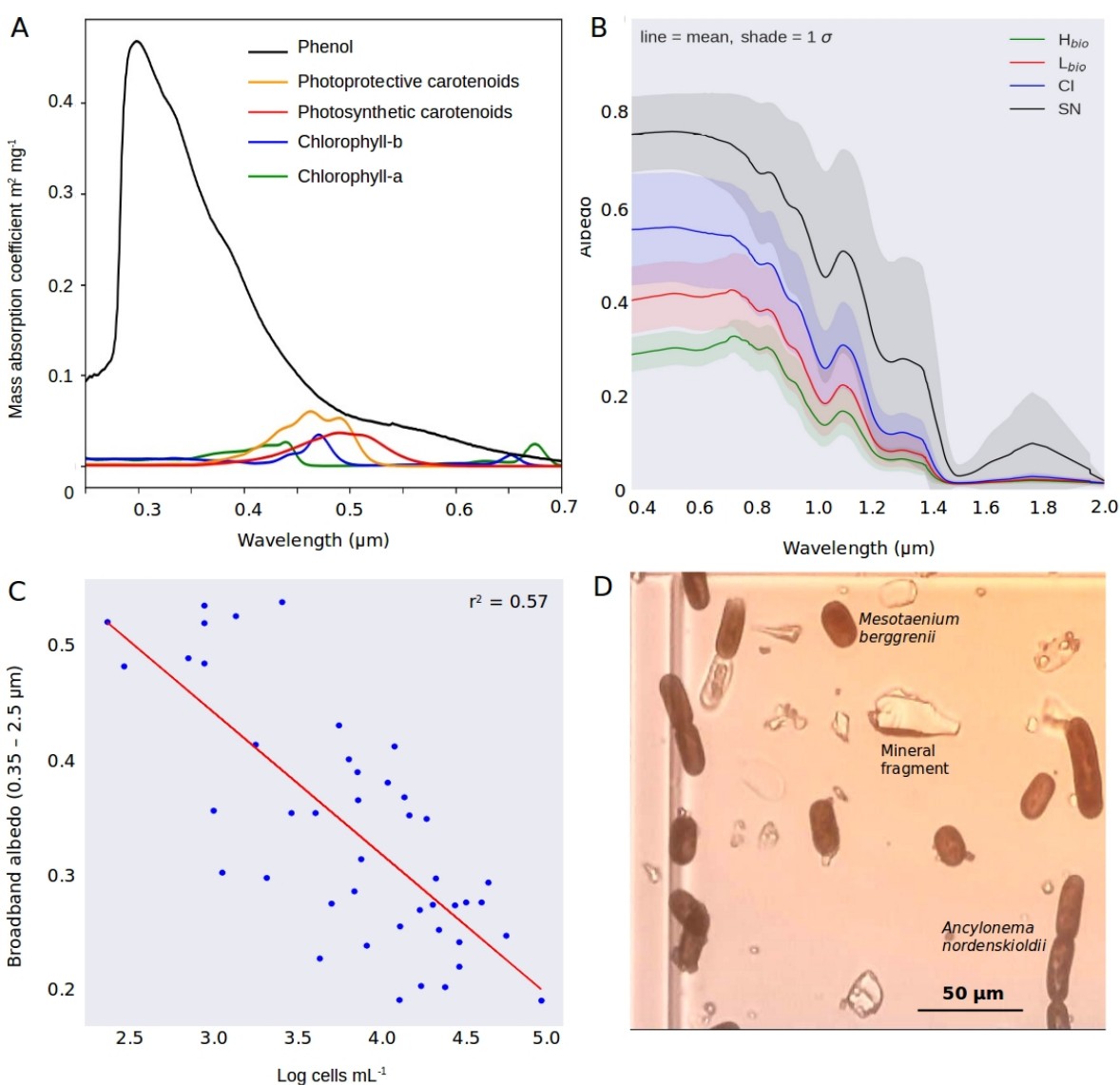


Fig 2 A) mass absorption coefficients of the major algal pigments including the purpurogallin-type phenol; B) Measured spectral albedos for each surface type, C) plot showing the natural logarithm of cell abundance against broadband albedo; D) microscope image showing examples of both algal species and mineral fragments from a melted $H_{bio}$ sample.


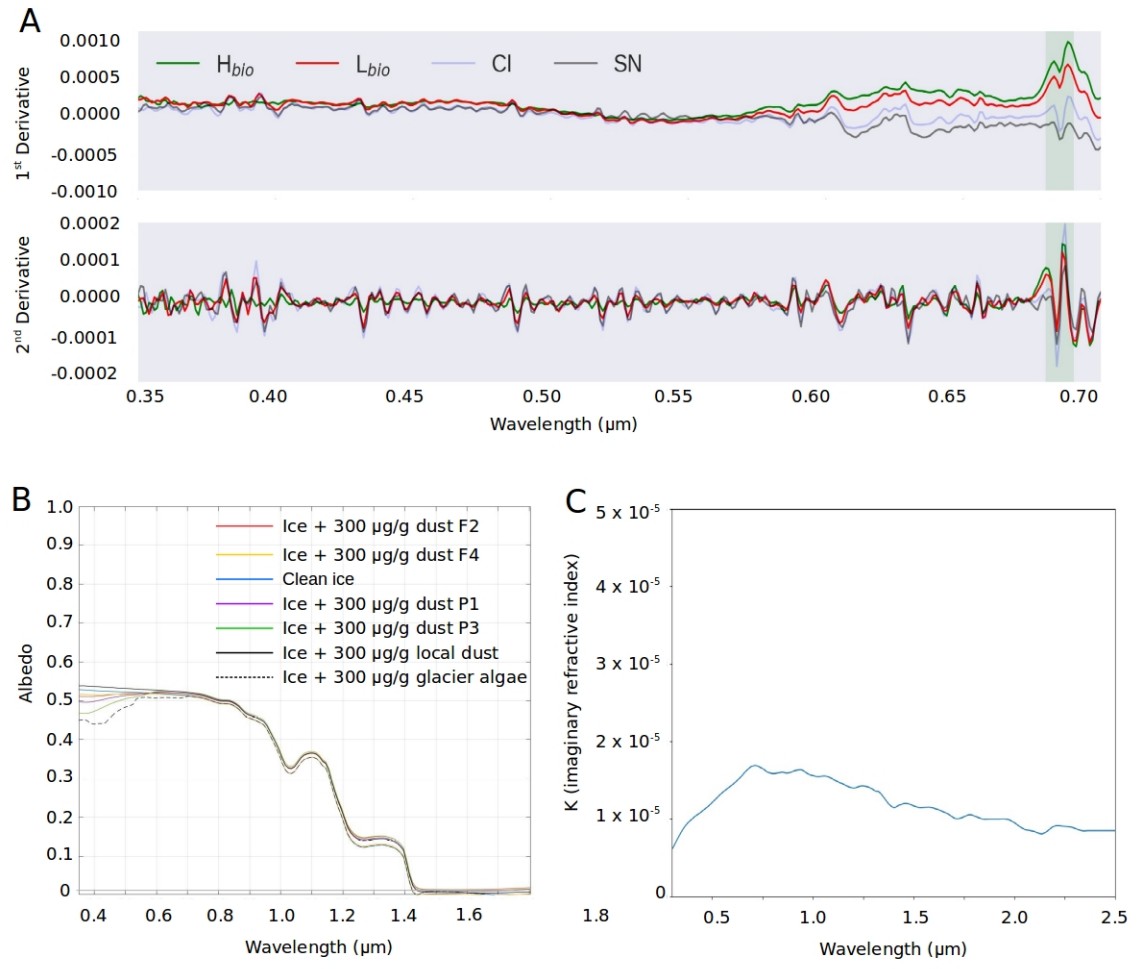

Figure 3: A) First and second derivative spectra for each surface class; B) BioSNICAR_GO modelled spectral albedo for clean ice (blue), and ice with 300 µg/g od a selection of mineral dusts (F2, F4, P1, P3, local) and algae in the upper 1 mm; C) Imaginary refractive index for the local mineral dust.


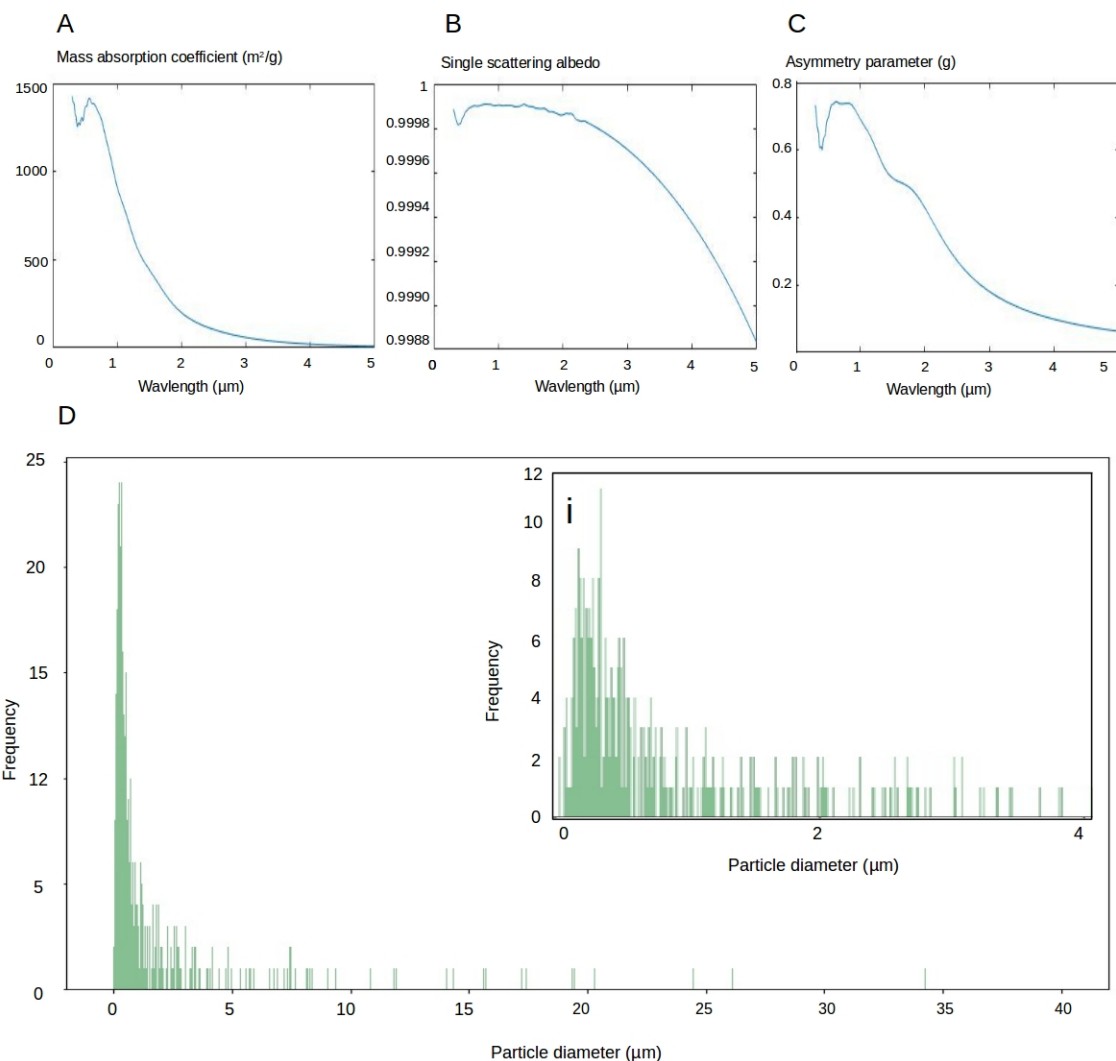


Figure 4: A) mass absorption coefficient calculated using Lorenz-Mie theory for the mineral dust sample, B) single scattering albedo calculated using Lorenz-Mie theory for the mineral dust sample, C) asymmetry parameter calculated using Lorenz-Mie theory for the mineral dust sample; D) Particle size diameter for our local mineral dust sample (inset i shows magnification of 0-4 µm range)





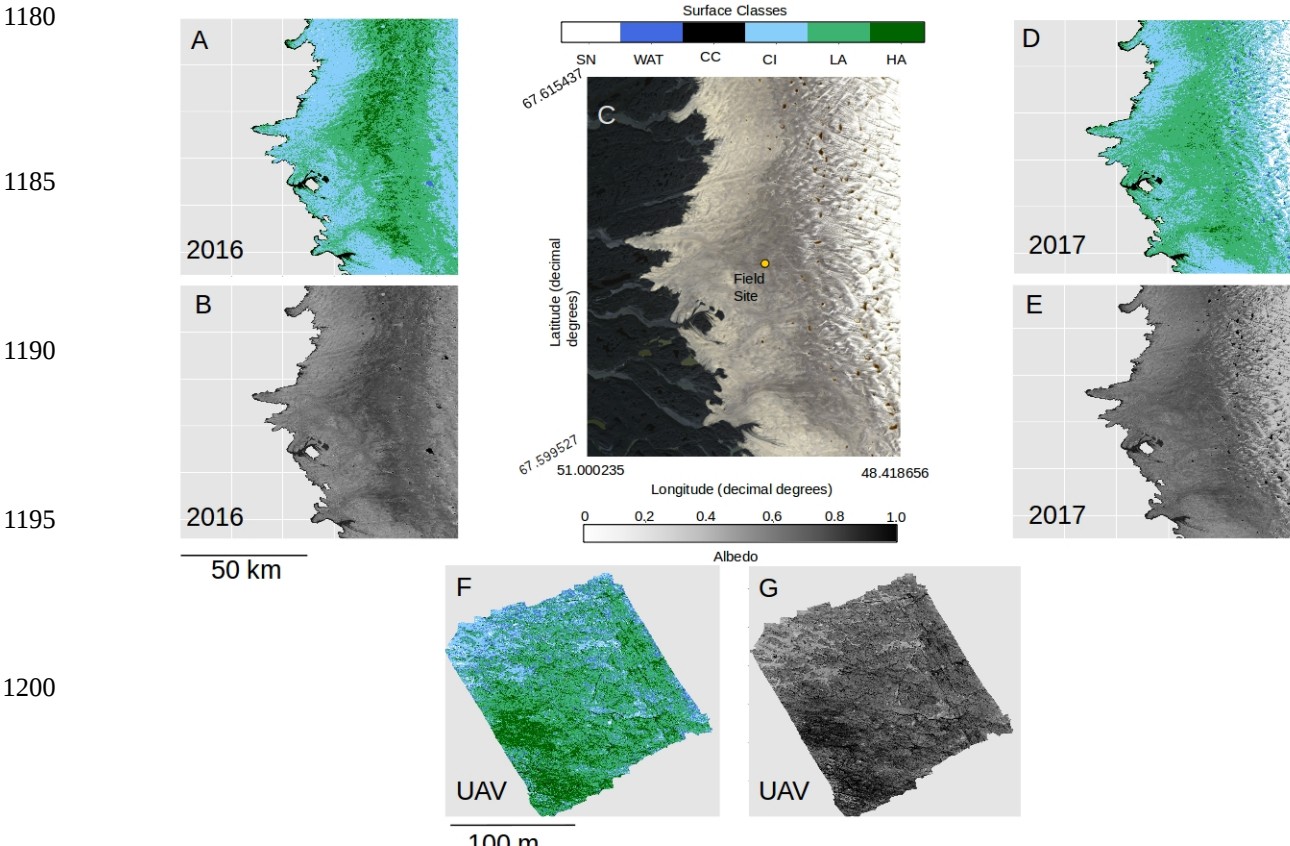

Fig 5: A) Classified map of the area shown in C for 2016. B) Broadband albedo map of the area shown in C for 2016. C) RGB "true colour" image showing the Sentinel 2 tile covering our field site in the Kangerlussuaq area. D) Classified map of the area shown in C for 2017. E) Broadband albedo map of the area shown in C for 2017. F) Classified map of a 200 x 200 m area at the field site marked in C imaged using a UAV mounted multispectral camera. G) Broadband albedo map of a 200 x 200 m area at the field site marked in C imaged using a UAV mounted multispectral camera. Panels A, B, C, D, E all use UTM Zone 22 projection and have pixel resolution of 20 m. The scale bare beneath panel B is common to panels A,B,D,E and the scale bar beneath panel F is common to panels F and G.

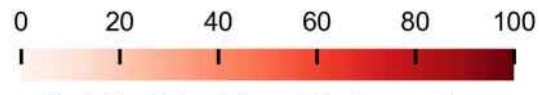

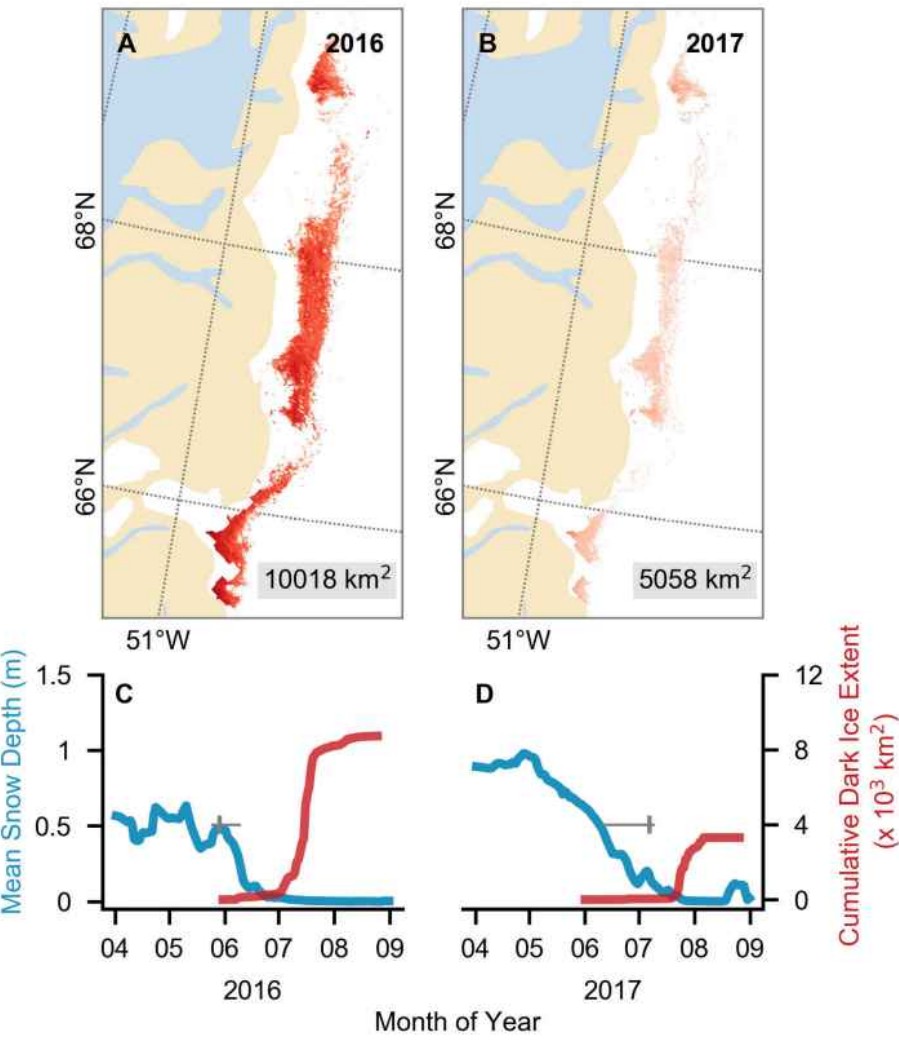

Figure 6: (A,B) Dark ice duration on the south-west GrIS in summers 2016 and 2017, expressed as a percentage of the total daily cloud-free observations made during June-July-August (JJA). Each year is labelled with dark ice extent. In each year, pixels that are dark for fewer than 5 days are not shown. (C,D) Average snow depth modelled by MAR (blue) and cumulative dark ice extent observed by MODIS (red) (Tedstone et al., 2017) during April to August. Vertical bars (grey) denote median date of snow clearing derived from MODIS; horizontal bars denote the interquartile range of the day of year of bare ice appearance. Tick marks denote the start of each month.


| 100 ug/g | Local dust | P1 | P2 | P3 | F1 | F2 | F3 | F4 | Glacier algae |
|---|---|---|---|---|---|---|---|---|---|
| Albedo change | +0.0025 | -0.0007.38 | -0.0016 | -0.0039 | +0.0032 | +0.000167 | -0.0012 | -0.0013 | -0.0065 |

| 300 ug/g | Local dust | P1 | P2 | P3 | F1 | F2 | F3 | F4 | Glacier algae |
|---|---|---|---|---|---|---|---|---|---|
| Albedo change | +0.0084 | -0.00054 | -0.0029 | -0.0092 | +0.011 | +0.0024 | -0.0024 | -0.0033 | -0.019 |

| 500 ug/g | Local dust | P1 | P2 | P3 | F1 | F2 | F3 | F4 | Glacier algae |
|---|---|---|---|---|---|---|---|---|---|
| Albedo change | +0.0145 | +0.000096 | -0.0033 | -0.0128 | +0.0185 | +0.0049 | -0.0031 | -0.0052 | -0.029 |

Table 1: Albedo change relative to clean ice resulting from the addition of 100, 300 and 500 ug/g of each mineral dust.







**A:**

| Surface Type | Mean | Standard deviation | Number of observations |
|---|---|---|---|
| WAT | 0.31 | 0.017 | 154070 |
| CC | 0.09 | 0.031 | 160448 |
| CI | 0.53 | 0.026 | 2735603 |
| $L_{bio}$ | 0.44 | 0.055 | 12098635 |
| $H_{bio}$ | 0.25 | 0.039 | 3447152 |
| SN | 0.74 | 0.025 | 63647 |

**B:**

| Surface Type | Mean | Standard deviation | Number of observations |
|---|---|---|---|
| WAT | 0.08 | 0.039 | 174791 |
| CC | 0.11 | 0.034 | 258520 |
| CI | 0.46 | 0.075 | 5947314 |
| $L_{bio}$ | 0.31 | 0.042 | 8740186 |
| $H_{bio}$ | 0.22 | 0.026 | 2270206 |
| SN | 0.76 | 0.058 | 16333853 |

**C:**

| Surface Type | Mean | Standard deviation | Number of observations |
|---|---|---|---|
| WAT | 0.08 | 0.044 | 52060 |
| CC | 0.13 | 0.035 | 272419 |
| CI | 0.46 | 0.042 | 6771763 |
| $L_{bio}$ | 0.32 | 0.046 | 8388680 |
| $H_{bio}$ | 0.23 | 0.028 | 1410095 |
| SN | 0.60 | 0.050 | 9924 |


**D:**

| Surface Type | Mean | Standard deviation | Number of observations |
|---|---|---|---|
| CI | 0.50 | 0.02 | 22 |

| | | | |
|---|---|---|---|
| L$_{bio}$ | 0.36 | 0.07 | 28 |
| H$_{bio}$ | 0.24 | 0.03 | 22 |
| SN | 0.56 | 0.10 | 5 |

Table 2: A) summary of the albedo for each surface class as predicted from our classified UAV image. B) summary of the albedo for each surface class as predicted from our classified Sentinel-2 image for 2017. C) summary of the albedo for each surface class as predicted from our classified Sentinel-2 image for 2016. D) summary of the broadband albedo for each surface class as measured using field spectroscopy at our field site in 2017 (we do not have cosine-collector albedo measurements for water or cryoconite surfaces).

| | UAV Image | Sentinel 2 (2016) | Sentinel 2 (2017) |
|---|---|---|---|
| **Total Image Area (km$^2$)** | 0.04 | 10,000 | 10,000 |
| **Total Algae (%)** | 78.5 | 57.99 | 58.87 |
| **H$_{bio}$ (%)** | 17.4 | 8.35 | 2.54 |
| **L$_{bio}$ (%)** | 61.08 | 49.65 | 56.33 |
| **Cryoconite (%)** | 0.82 | 1.61 | 1.67 |
| **Clean Ice (%)** | 13.81 | 40.08 | 38.34 |
| **Water (%)** | 0.78 | 0.31 | 1.13 |
| **Snow (%)** | 6.09 | n/a | n/a |

1335

Table 3: percentage of each image covered by each surface type as predicted by our trained RF algorithm. Snow was removed from the calculation in the Sentinel-2 images to enable quantification of surface coverage in the bare-ice zone, i.e. below the snow line, only.

1340

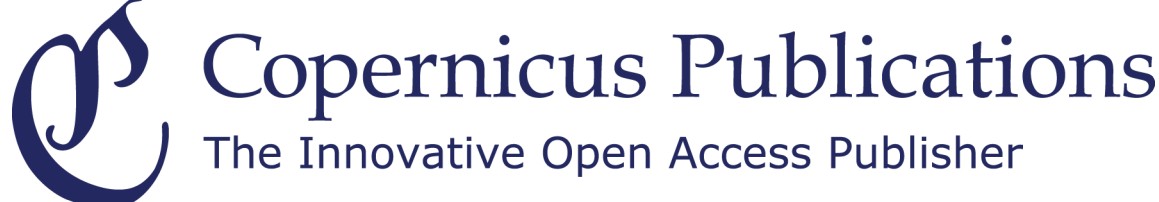

**Figure 1: The logo of Copernicus Publications.**