# Peer review of "Glacier algae accelerate melt rates on the south-western Greenland Ice Sheet"

_The Cryosphere, 2019_

## Referee Comment (RC1) · Anonymous Referee #1 · 5 May 2019

The study by Cook et al. aims to quantify the impact of glacier algae on surface melt and runoff across the western Greenland Ice Sheet by combining the field observations, radiative transfer modeling, remote sensing classifications, and surface runoff modeling. The topic of this study is important and interesting to the cryosphere community. However, some issues need to be addressed before publication.

1) The title is too general, given that the algae observations are from a single field site and the surface runoff estimates are based on a transect with forcing data from three automatic weather stations. After reading the whole manuscript, I find that it is not so convincing that the field observations of this site are representative for the entire Greenland Ice Sheet.

2) The estimation of surface runoff caused by algae over the western GrIS is not clearly

presented. Did you use the classification results from UAV and Sentinel-2 to estimate the surface runoff over the entire western GrIS? How? Your sentinel-2 and UAV imagery cover a very small portion of the western GrIS. How did you generalize the results? It seems that you only modeled the surface runoff over three points along a transect where forcing data from weather stations are available, and then extrapolated the runoff estimates across the whole area based on elevations. I believe there are lots of uncertainties here, even without considering the spatial heterogeneity of surface albedo. So in your abstract where you concluded that 'algal growth led to an additional 5.5-8.0 Gt of runoff from the western sector of the GrIS in summer 2016', an uncertainty estimate is mandatory.

3) Cook et al. attributed the albedo reduction to glacier algae because mineral dust was considered as less effective on albedo reduction based on the radiative transfer modeling. However, the surface meltwater itself has a significant role in reducing the albedo, which was not considered and evaluated.

4) Many critical details on methods and results are missing, which need clarifications. See specific comments below.

5) The overall writing needs to be improved, particularly the writing style. The method and result parts are poorly structured, which seem like a simple but loose stacking of various materials, while the logical linkages between different parts are weak and not clear. This study involves several different components, including fieldwork, radiative transfer modeling, image classification from UAV and Sentinel-2 data, and surface runoff modeling using 'an SMB model.' In the second section 'Field sites and methods,' all the materials related to those components are just put together, which are very difficult for readers to follow. There are also many redundant descriptions between the method and result parts, which seems like that this manuscript was not thoroughly proofread. There are also some grammar errors and typos.

6) The most recent literature about ice algae mapping using remote sensing data is not

cited and discussed, like:

Wang, S., Tedesco, M., Xu, M., & Alexander, P. M. (2018). Mapping Ice Algal Blooms in Southwest Greenland From Space. Geophysical Research Letters, 45(21), 11,779–11,788.

Huovinen, P., Ramírez, J., & Gómez, I. (2018). Remote sensing of albedo-reducing snow algae and impurities in the Maritime Antarctica. ISPRS Journal of Photogrammetry and Remote Sensing: Official Publication of the International Society for Photogrammetry and Remote Sensing, 146, 507–517.

Both these two papers are using remote sensing data to detect snow/ice algae. Although the second paper is dealing with snow algae, the first paper is utilizing the chlorophyll-a signature to map ice algae from satellite imagery over the southwestern GrIS. Your presented field data and radiative transfer modeling results (particularly your Figure 2A) are consistent with Wang et al. (2018) who used the reflectance ratio between 709 and 673 nm to quantify the ice algae.

Specific comments.

Introduction: This section should be expanded, at least including a more detailed literature review about the current research progress and efforts on ice algae and their relationship with albedo and surface melting.

Line 62. The study by Wang et al. (2018) should be cited, which used the spectral signature of chlorophyll-a to map ice algae in Greenland.

Line 78. Adjust your figures. It's a bit odd to put your first figure in the text as Figure 3c.

Line 86. Do you have multiple sites? What's your sampling area size? Did you take point measurements at different places within a specific area? Explain those details in your field site description part.

Line 99. Update your reference 'Cook et al. (2017b)' in the reference list.
2.3 Biological Measurements. Did you also differentiate different species when counting the cells?

Line 126. What is PSD? Particle size distribution? Do not use abbreviation when you use the term first time in your manuscript.

Line 135. Provide more details on how you used the ASD to measure the surface reflectance of the materials pressed on the microscope slide, such as your measurement setup, the field view of the ASD probe, and the background material (white, grey, or black) where you put your microscope slide.

Line 144. Any references for the BioSNICAR_GO? Is BioSNICAR developed for this study? Provide more details.

Section 2.5 paragraph 2. As I asked before if you considered the different cell numbers of different algal species, did you take into account the different shapes of the Ancylonema nordenskiöldii, Mesotaenium berggrenii? Ancylonema nordenskiöldii is filamentous while Mesotaenium berggrenii is unicellular. If you consider the geometrical optics, how would these two different shapes affect the radiative transfer modeling? Can you comment on the sensitivity of radiative transfer modeling on algal cell shapes?

Can you combine sections 2.5 and 2.6?

Line 187. 'utilise 5% of this ...' any references for this?

Line 191. determine -> determined

Section 2.7 paragraph 2. This paragraph is overall difficult to read. Could you use some equations to show your calculations?

2.8 UAV remote sensing. When did you conduct the UAV mapping, the same time with your field spectral collection? Also specify the multispectral camera parameters, like band wavelength, bandwidth and so on.

Line 210. oin -> in

Line 216. What do you mean by time-dependent regression?

Line 222. What do you mean by 'generally good'? Move this to the results or discussion part and make clarifications.

Section 2.9. It's very odd to have just two sentences in a single section. You should combine this with classification, or introduce more about the Sentinel-2 data. How does the Sen2Cor perform? Can you provide a figure showing the atmospherically corrected surface reflectances of Sentinel-2 data? What are the acquisition dates of your Sentinel-2 data?

Line 231. remove the word 'novel' as random forest classification has been widely used.

Line 234. What tiles? Sentinel-2?

Line 236. How did you reduce the ASD spectra to the UAV bands considering the difference between their bandwidth? Please clarify.

Line 240. This part is not clear. The reflectances at five wavelengths (reduced from ASD spectra?) were used as the feature vector, what's your classification vector? How many classes and what classes you were training? How many training samples do you have?

Line 269. clarify the 'rolling-window approach' or use reference

Line 271-276. Rewrite this part. Specify your surface class.

Line 283. 'surface albedo is adjusted from MODIS...' How?

2.12 Runoff modeling. What's the relationship between your remote sensing classification and runoff modeling. Did you use the classification results to constrain your runoff modeling? Uncertainty estimation should be included. The surface runoff estimation is not rigorous considering the albedo difference.

Line 301-307. Combine these texts with your field site and measurements description in section 2.

Line 312-313: Does the number after '+/-' mean standard deviation? Change the symbol to '$\pm$'. What standards did you take to separate the samples into those four classes? Can you show the histogram of your samples and the separating boundaries of the classes?

Line 360. Plot the absorption spectrum for the purpurogallin pigment, with other photosynthetic and photoprotective pigments.

Line 374. Be cautious about making this conclusion based on your field measurements over just one specific area.

Line 383-385. You didn't take any biological measurements over those 'wavy' areas. Generalizing your single-site observation to the entire GrIS is inappropriate.

Section 3.5. This part needs to be rewritten. Many texts should go to the methods section. I find that this manuscript has a lot of those redundant descriptions. Some texts should be in the previous section but were put in the results section. Can you plot out the spectra (reduced to UAV and Sentinel-2 bands) of the four different classes, in comparison with your original ASD spectra? The ASD spectra may well differentiate four different classes, but the reduced spectra would mask out lots of unique spectral signatures considering the coarse spectral resolutions. Otherwise, your classification (no matter what advanced methods to be used) is not supported. Besides, you should also plot out the real UAV and Sentinel-2 spectra, in comparison with your ASD spectra.

Line 444. The this -> The.

Section 3.8. Discuss the potential impact of meltwater itself.

Table 1. explain the abbreviation in your table title or put a note on this.

---

## Referee Comment (RC2) · Stephen Warren (Referee) · 6 Jun 2019

Review for TCD by Stephen Warren, June 2019
Cook et al.: "Glacier algae accelerate melt rates on the western Greenland Ice Sheet".
Recommendation:  major revision required.

**General statement**
This is an interesting and important paper, attempting to quantify the role of algal growth in the energy budget of the Greenland Ice Sheet during the melting season. By comparing and contrasting two dramatically different summer seasons, the importance of interannual variability is shown. The paper should be published after the major comments are addressed.

**Major comments**

(1) My main concern, the one requiring "major revision", is the inference of optical properties for dust. Figure 2B shows the imaginary part of the refractive index increasing by a factor of 4.4 from 0.4 to 0.7 µm wavelength. This contrasts with the usual finding for desert dust, whose imaginary index *decreases* by a factor of about 4 from 0.4 to 0.7 µm, resulting in a red color in reflection (e.g. Patterson et al. 1977, Müller et al. 2009, Wagner et al. 2012). [These and other references are discussed by Dang et al. (2015, Section 6.4).] The dust of Figure 2B would have a blue color.  Blue minerals do exist, but they are rare.  I am therefore skeptical that Skiles's method is obtaining the correct answer.  If the absorption spectrum of Figure 2B is to be believed, the authors should present evidence for blue minerals in dust that deposits in this part of Greenland.  Furthermore, the imaginary index shown for wavelength 0.5 µm ($4 \times 10^{-6}$) is a factor of 1000 smaller than what is typical of desert dust in the references cited above.

Figure 2C shows the albedo effect of an arbitrary 300 ppm of dust. I didn't find the actual dust concentration given in the paper for the ice that was measured; maybe I missed it. In any case, a plot or table of the albedo effect of the measured dust concentration should be added to the paper.

(2) The size distribution for dust is shown in a supplementary figure 1E, but the interesting data are crowded up to the far left end of the figure, so the location and shape of the peak are hard to see. I would like to see an expansion of the 0-3 µm region of this figure. This is an important finding, so the new figure should be put in the main paper rather than in the supplement.

(3) Color of algae.  The algae in Figure 1B are bright red.  But the dark zone in Figure 1A is gray, not red.  Please explain.

(4) The ice surface is modelled to consist of large hexagonal columns. That is unlikely to be realistic; the shapes are probably irregular. The main thing you need to consider is what is responsible for scattering of light in glacier ice, namely bubbles and cracks, which together determine the most important quantity, namely the specific surface area (SSA). Then a simpler radiative-transfer method could be used, as was done for glacier ice by Dadic et al. (2013, Section 6.1).

(5) The authors collected data from a "dark" year (2016) and a "bright" year (2017), providing an interesting contrast.  But the abstract and the conclusion, which should summarize the results, instead give the results only for the dark year, thus exaggerating the average effect of the algae.

The reader will thus conclude that the authors are claiming more importance for their research topic than is justified, which is a shame. The authors have missed the opportunity to use the 2016-2017 contrast to highlight the effect of future climatic expansion of the snow-free season.

(6) It would be good to extend this analysis to cover all of the GrIS. Is that possible, or do the Sentinel satellites not survey the entire ice sheet?

(7) The author-list may be too long. The Author Contribution Statement concludes with the statement: "Other authors commented on the style and content of the final manuscript." This contribution alone does not qualify one to be an author, according to established principles of authorship, as given for example at
http://www.icmje.org/recommendations/browse/roles-and-responsibilities/defining-the-role-of-authors-and-contributors.html
Dr. Cook should carefully consider whether some of his 23 coauthors should be moved to the Acknowledgments. By crowding into the author-list, they deprive the real workers of the credit they deserve.

**Minor comments**
line 34. Change "albedo-reducing" to "albedo-enhancing" (since addition of dust raised the albedo).
line 35. "western sector of the GrIS" Define, by adding the span of latitudes (65-70 N).
line 187. Give a reference for photosynthesis using 5% of absorbed sunlight.
line 191 (also 425). Latent heat of fusion is 334 J/g, not 334 J/cm$^3$. This is important since on line 75 the ice density is varied from 0.3 to 0.7 g/cm$^3$.
line 288. Give a reference for these roughness-heights.
line 313. "SN=0". I am surprised that there was no algae in the snow.
line 400. Define what you mean by "area" of an absorption feature.
line 440-441. "Directional reflectance . . . approximates the measurements made by orbital remote sensing platforms". This is true if the satellite is nadir-viewing, since your surface measurement was a nadir view. Otherwise, your surface measurement is biased low because you are looking directly down into cryoconite holes, whereas the cryoconite material is hidden from the view of an obliquely-viewing satellite.
line 550-551. "We demonstrate that the growth of algae occurs over a large proportion of the ablating area of the GrIS". For this statement to be true, you would need to give estimates for the entire ablation zone, not just latitudes 65-70 of West Greenland.
Figure 3 caption. On line 1, "Albedo map" is for what wavelength? On line 3, change "D" to "E", change "E" to "F". On line 4, change "F" to "G".
Figure 4C,D. The tick marks for months are for the beginning of the month or the middle of the month?
Figure 4D, vertical axis label. What does the adjective "cumulative" mean? Normally "cumulative" implies an integral.
Table 2. The percentages for the UAV add to 94%. What is the remaining 6%? For the Sentinel columns, the percentages add to 1%; probably they should all be multiplied by 100.
Figure S1A, vertical axis label. Define IRF.
Figure S1B. Change the color coding to agree with Figure 1C.
Figure S1 caption, line 1-2. Delete "per unit wavelength", because the values are just unitless.

**Very minor comments**
line 64.  Ryan et al 2018.  Do you mean 2018a or 2018b?
line 76.  Define IMAU.
line 107.  Define GNU.
line 109.  Hildebrand.  The reference list says instead Hillebrand.
line 127.  Define PSD.
line 152.  Cook et al 2017.  Do you mean 2017a or 2017b?
line 160.  Dauchet et al. 2015 is missing from the reference list.
line 169.  Warren and Brandt 2008 is missing from the reference list.
line 180.  Define IRF.
line 362.  Lee and Pilon 2013 is missing from the reference list.
lines 535-537.  Say that this sentence applies to the year 2016.
line 614.  Update reference from TCD to TC.
line 620.  Reference is out of order.
line 739.  Reference is out of order.
line 743.  Update reference from TCD to TC.

**Spelling and punctuation**

*Hyphenate these adjectives:*
albedo-reducing (lines 31, 61, 339, 343, 347)
light-absorbing (lines 48, 69, 324, 531)
bare-ice (line 63, 263, 269)
long-term (line 65)
ice-sheet (line 67, 231, 389)
remote-sensing (line 72, 78, 441)
high-algal-biomass (line 115)
volume-weighted (line 152)
satellite-derived (line 222)
random-forest (line 232)
cloud-free (line 235, 272)
dark-ice (line 265, 266)
two-stream (line 346)
low-albedo (line 496)

line 87.  Insert comma after "conditions"
line 191. Change "determine" to "determined".
line 210. "oin".  You probably mean "in" or "on".
line 238.  Insert comma after "columns".
line 256.  Change "model are" to "models are".
line 262.  Insert comma after "Terra".
line 275.  Insert comma after "year".
line 276.  Insert comma after "available"
line 441.  Change "are" to "is"
line 443.  "The this" needs fixing.
line 645.  Capitalise "smith"

line 658.  Fix the author-list.
line 678.  Change Fettweiss to Fettweis
line 714.  Fix "H.-G.rensen"
line 746.  Fix "sheetL"
Supp Info 3. Change "Assymetry" to "Asymmetry" in three places.

**References**
Dadic, R., P.C. Mullen, M. Schneebeli, R.E. Brandt, and S.G. Warren, 2013:  Effects of bubbles, cracks, and volcanic tephra on the spectral albedo of bare ice near the Trans-Antarctic Mountains:  implications for sea-glaciers on Snowball Earth. *J. Geophys. Res. (Earth Surfaces)*, 118, doi:10.1002/jgrf.20098.

Dang, C., R.E. Brandt, and S.G. Warren, 2015:  Parameterizations for narrowband and broadband albedo of pure snow, and snow containing mineral dust and black carbon.  *J. Geophys. Res.*, 120, doi:10.1002/2014JD022646.

Müller, T., A. Schladitz, A. Massling, N. Kaaden, K. Kandler, and A. Wiedensohler (2009), Spectral absorption coefficients and imaginary parts of refractive indices of Saharan dust during SAMUM-1, *Tellus B, 61*, 79–95.

Patterson, E. M., D. A. Gillette and B. H. Stockton (1977), Complex index of refraction between 300 and 700 nm for Saharan aerosols, *J. Geophys. Res., 82*, 3153-3160.

Wagner, R., T. Ajtai, K. Kandler, K. Lieke, C. Linke, T. Müller, M. Schnaiter, and M. Vragel (2012), Complex refractive indices of Saharan dust samples at visible and near UV wavelengths:  a laboratory study, *Atmos. Chem. Phys., 12,* 2491-2512.

---

## Author Comment (AC1) · 4 Jul 2019

**Cook et al.: "Glacier algae accelerate melt rates on the western Greenland Ice Sheet".**
**Recommendation: major revision required.**

*We are very grateful for the reviewer's critique of our paper. Several important points were raised that we have reflected upon and endeavoured to address in the revised manuscript. We have made substantial revisions that we hope have improved the manuscript. In particular, we have repeated our mineral imaginary refractive index retrieval and also added a sensitivity study showing the response of glacier ice-albedo to a range of Saharan dusts. We have also restructured the paper to better demonstrate the potential interannual feedback. Detailed responses to the major and minor comments are detailed below. Please note that we have uploaded one revised manuscript to satisfy both sets of review comments.*

**General statement**

This is an interesting and important paper, attempting to quantify the role of algal growth in the energy budget of the Greenland Ice Sheet during the melting season. By comparing and contrasting two dramatically different summer seasons, the importance of interannual variability is shown. The paper should be published after the major comments are addressed.

**Major comments**

(1) My main concern, the one requiring "major revision", is the inference of optical properties for dust. Figure 2B shows the imaginary part of the refractive index increasing by a factor of 4.4 from 0.4 to 0.7 ʎm wavelength. This contrasts with the usual finding for desert dust, whose imaginary index decreases by a factor of about 4 from 0.4 to 0.7 um, resulting in a red color in reflection (e.g. Patterson et al. 1977, Müller et al. 2009, Wagner et al. 2012). [These and other references are discussed by Dang et al. (2015, Section 6.4).] The dust of Figure 2B would have a blue color. Blue minerals do exist, but they are rare. I am therefore skeptical that Skiles's method is obtaining the correct answer. If the absorption spectrum of Figure 2B is to be believed, the authors should present evidence for blue minerals in dust that deposits in this part of Greenland. Furthermore, the imaginary index shown for wavelength 0.5 um (4×10 -6 ) is a factor of 1000 smaller than what is typical of desert dust in the references cited above. Figure 2C shows the albedo effect of an arbitrary 300 ppm of dust. I didn't find the actual dust concentration given in the paper for the ice that was measured; maybe I missed it. In any case, a plot or table of the albedo effect of the measured dust concentration should be added to the paper.

*We thank the reviewer for pointing this out – it is correct to point out the unrealistic refractive index. Thanks to the comment, we revisited our reflectance measurements and found that the reflectance measured in the integrating sphere did indeed suggest higher reflectance in the red wavelengths, which led us to a bug in the inverted DISORT retrieval. This was fixed and the refractive index revised and incorporated into our radiative transfer model. The single scattering optical properties are comparable to dust of a similar size range presented by Flanner et al. (2007) and Skiles et al. (2017). The dusts found on the bare ice are not characteristically Saharan, have only trace amounts of red minerals and are much less strongly absorbing than those in Wagner et al., Polashenski et al., etc. This is a consistent finding between our mineralogical investigations (which will comprise an entire paper to be submitted imminently by co-author McCutcheon) and past studies such as Wientjes et al. (2011) and Tedesco et al. (2013) on the K-transect and Sanna and Romeo (2018) slightly further north at Equip Sermia.*

*To protect against possible underestimation of the mineral albedo-lowering effect, we also gathered optical properties for other dusts relevant to our study and included those in the model too, enabling some sensitivity testing. Those other dusts are the "global average" dusts from Flanner et al. (2007) and those from Polashenski et al. (2015) who recreated GrIS dusts from past literature studies and simulated low, median and high hematite scenarios. Those dusts are "characteristically Saharan" so as well as providing a sensitivity test, they also demonstrate the effects of possible future Saharan dust deposits by atmospheric transport. In the revised manuscript we compared the albedo reducing effects of algae against our real mineral dusts as well as those from the literature.*

*With our new k values and sensitivity study, our conclusion that algae are the dominant light absorbing particle remains unchanged. Please see the revised manuscript for full details of the analysis and findings.*

(2) The size distribution for dust is shown in a supplementary figure 1E, but the interesting data are crowded up to the far left end of the figure, so the location and shape of the peak are hard to see. I would like to see an expansion of the 0-3 um region of this figure. This is an important finding, so the new figure should be put in the main paper rather than in the supplement.

*We have provided an updated PSD plot with an inset frame showing the range 0-4 um. This is presented along with the mineral dust single scattering optical properties in Figure 4 in the revised manuscript.*

(3) Color of algae. The algae in Figure 1B are bright red. But the dark zone in Figure 1A is gray, not red. Please explain.

*We agree - the microscope image is too red and does not represent the true colour of the algae – it was selected because it clearly shows the cell morphology, but on reflection the colour is critical to portray accurately. A new image has been added that more accurately shows the true colour of the algae. Sometimes the algae really do have a red-purple tinge to them that can even be discerned by eye in localised areas of very heavy biomass loading.*

(4) The ice surface is modelled to consist of large hexagonal columns. That is unlikely to be realistic; the shapes are probably irregular. The main thing you need to consider is what is responsible for scattering of light in glacier ice, namely bubbles and cracks, which together determine the most important quantity, namely the specific surface area (SSA). Then a simpler radiative-transfer method could be used, as was done for glacier ice by Dadic et al. (2013, Section 6.1).

*The near surface is composed of large, irregularly shaped grains as the reviewer suggests, and the SSA is the primary factor governing scattering in these ice media. To my knowledge, the assumption of hexagonally shaped grains shouldn't invalidate our approach as long as the SSA of the upper layers of ice is realistic. Since we do not have a good method for determining the SSA of the weathered layer of bare-ice empirically, we can only judge this by how closely we can recreate our field spectra. The new model does a much better job of this than the original SNICAR model based on Mie scattering. I believe that the new model is the best available for ice algae on weathered ice and a useful addition to the literature, but I do also appreciate its limitations and would be very interested to follow up by more closely examining the Dadic et al. (2013) model. We have added a sentence to acknowledge this in section 2.5.*

(5) The authors collected data from a "dark" year (2016) and a "bright" year (2017), providing an interesting contrast. But the abstract and the conclusion, which should summarize the results, instead give the results only for the dark year, thus exaggerating the average effect of the algae. The reader will thus conclude that the authors are claiming more importance for their research topic than is justified, which is a shame. The authors have missed the opportunity to use the 2016-2017 contrast to highlight the effect of future climatic expansion of the snow-free season.

*We agree fully and have reorganised and refined the manuscript to a) avoid exaggerating the effect of the algae, and b) use the interannual variability to demonstrate the potential growth-melt feedback.*

(6) It would be good to extend this analysis to cover all of the GrIS. Is that possible, or do the Sentinel satellites not survey the entire ice sheet?

*The classifier was deliberately not applied over the entire ice sheet because our lack of field samples from outside of the Kangerlussuaq region was prohibitive – before upscaling to the ice sheet scale it would be better to have ground spectra and samples from further north that indicate the same processes are in operation across the whole ablation zone. This is especially the case after addressing the comments from Reviewer 1, who had concerns about upscaling over the south-western region. However, scaling our classifier over the entire GrIS western margin for the entire Sentinel-2 record is underway, to be worked up into a paper once our field samples from near Upernavik (NW Greenland) are fully analysed.*

(7) The author-list may be too long. The Author Contribution Statement concludes with the statement: "Other authors commented on the style and content of the final manuscript." This contribution alone does not qualify one to be an author, according to established principles of authorship, as given for example at http://www.icmje.org/recommendations/browse/roles-and-responsibilities/defining-the-role-of-authors-and-contributors.html Dr. Cook should carefully consider whether some of his 23 coauthors should be moved to the Acknowledgments. By crowding into the author-list, they deprive the real workers of the credit they deserve.

*This paper represents the culmination of almost 3 years of dedicated work, drawing upon collaborations across a wide range of disciplines - many people have made important contributions along the way. The author statement has been amended to better reflect the contribution of these authors to the study.*

**Minor comments**

line 34. Change "albedo-reducing" to "albedo-enhancing" (since addition of dust raised the albedo).

*This has been addressed by default since we have updated our whole mineralogy section*

line 35. "western sector of the GrIS" Define, by adding the span of latitudes (65-70 N).

*This has been amended*

line 187. Give a reference for photosynthesis using 5% of absorbed sunlight.

*Citations added*

line 191 (also 425). Latent heat of fusion is 334 J/g, not 334 J/cm 3 . This is important since on line 75 the ice density is varied from 0.3 to 0.7 g/cm 3 .

*This has been amended*

line 288. Give a reference for these roughness-heights.

*Citations added*

line 313. "SN=0". I am surprised that there was no algae in the snow.

*We also expected that a snow algal bloom might form on the melting snow, and this was part of the motivation for establishing a field camp above the snow line in 2017. However, we never observed snow algae, even in light patches, forming on the snow at any point during the transition from dry snow to slush to bare-ice exposure, and this is corroborated by our field samples. We sometimes found them in samples of cryoconite, and we know from previous molecular work (Lutz et al. 2018) they are present in the environment, but we have not yet witnessed snow algae taxa in microscope samples or by visual assessment of the ice surface at any time across three field seasons between 2016 – 18 period, apart from one instance of a red algal bloom seen on snow inside a crevasse near the ice margin in August 2017, which we saw from the helicopter window as we departed!*

line 400. Define what you mean by "area" of an absorption feature.

*This has now been defined in the manuscript*

line 440-441. "Directional reflectance . . . approximates the measurements made by orbital remote sensing platforms". This is true if the satellite is nadir-viewing, since your surface measurement was a nadir view. Otherwise, your surface measurement is biased low because you are looking directly down into cryoconite holes, whereas the cryoconite material is hidden from the view of an obliquely-viewing satellite.

*Yes, there may be some uncertainty due to off-nadir satellite viewing angles; however it is still a more appropriate measurement than albedo measured using a cosine collector.*

line 550-551. "We demonstrate that the growth of algae occurs over a large proportion of the ablating area of the GrIS". For this statement to be true, you would need to give estimates for the entire ablation zone, not just latitudes 65-70 of West Greenland.

*This has been amended*

Figure 3 caption. On line 1, "Albedo map" is for what wavelength? On line 3, change "D" to "E", change "E" to "F". On line 4, change "F" to "G".

*Thank you for pointing out these errors, we have amended them. "Albedo" has been changed to "Broadband albedo" - the values were for the whole solar spectrum calculated from multispectral reflectance using the Liang et al. (2000) or Knap et al (1999) narrowband to broadband conversion equations.*

Figure 4C,D. The tick marks for months are for the beginning of the month or the middle of the month?

*The tick marks are for the beginning of the month – this is now stated in the figure caption.*

Figure 4D, vertical axis label. What does the adjective "cumulative" mean? Normally "cumulative" implies an integral.

*Cumulative refers to the running total, i.e. the cumulative sum of pixels that have gone "dark" over the ablation period.*

Table 2. The percentages for the UAV add to 94%. What is the remaining 6%? For the Sentinel columns, the percentages add to 1%; probably they should all be multiplied by 100.

*The remainder was classified as snow. We did not report the snow because we are interested in the bare ice zone, and the snow detected in S2 imagery was at or above the snow line, skewing our proportional coverage calculations for the bare ice classes. On reflection, it is probably better to report the snow coverage for the UAV images, as there are some small rotten ice patches. These are not detected in the S2 imagery because of their very small spatial extent compared to the ground resolution of the S2 sensor. We have added the UAV snow coverage to the table and clarified in the table caption.*

Figure S1A, vertical axis label. Define IRF.

*This has been amended*

Figure S1B. Change the color coding to agree with Figure 1C.

*This has been amended*

Figure S1 caption, line 1-2. Delete "per unit wavelength", because the values are just unitless.

*This has been amended*

**Very minor comments**

line 64. Ryan et al 2018. Do you mean 2018a or 2018b?

*amended*

line 76. Define IMAU.

*Defined*

line 107. Define GNU.

*This is a recursive acronym nested within a recursive acronym…! GIMP stands for GNU Image Manipulation Program, where GNU itself is a recursive acronym that stands for "GNU Not Unix" and describes a collection of free operating system distributions based on UNIX - it might not be that helpful to define in the paper as the definition is GNU's all the way down…!*

line 109. Hildebrand. The reference list says instead Hillebrand.

*This has been corrected*

line 127. Define PSD.

*PSD has now been defined on first use.*

line 152. Cook et al 2017. Do you mean 2017a or 2017b?

*This has been clarified*

line 160. Dauchet et al. 2015 is missing from the reference list.

*Reference added*

line 169. Warren and Brandt 2008 is missing from the reference list.

*Reference added*

line 180. Define IRF.

*Now included in revised manuscript*

line 362. Lee and Pilon 2013 is missing from the reference list.

*Reference added*

lines 535-537. Say that this sentence applies to the year 2016.

*This sentence was revised in the new manuscript*

line 614. Update reference from TCD to TC.

*Amended*

line 620. Reference is out of order.

*Amended*

line 739. Reference is out of order.

*Amended*

line 743. Update reference from TCD to TC.

*Amended*

**Spelling and punctuation**
**Hyphenate these adjectives:**

albedo-reducing (lines 31, 61, 339, 343, 347): *amended throughout*
light-absorbing (lines 48, 69, 324, 531): *amended throughout*
bare-ice (line 63, 263, 269): *amended throughout*
long-term (line 65): *amended throughout*
ice-sheet (line 67, 231, 389): *amended throughout*
remote-sensing (line 72, 78, 441): *amended throughout*
high-algal-biomass (line 115): *removed during rewrite*
volume-weighted (line 152): *amended*
satellite-derived (line 222): *amended*
random-forest (line 232): *amended throughout*
cloud-free (line 235, 272): *Amended once, second instance removed in rewrite*
dark-ice (line 265, 266): *amended*
two-stream (line 346): *amended*
low-albedo (line 496): *amended*

line 87. Insert comma after "conditions": *comma added*
line 191. Change "determine" to "determined": *sentence removed in revised manuscript*
line 210. "oin". You probably mean "in" or "on": a*mended*
line 238. Insert comma after "columns".: *comma added*
line 256. Change "model are" to "models are".: *sentence removed in revised manuscript*
line 262. Insert comma after "Terra": *sentence removed in revised manuscript*
line 275. Insert comma after "year": *sentence removed in revised manuscript*
line 276. Insert comma after "available": *sentence removed in revised manuscript*
line 441. Change "are" to "is": *sentence removed in revised manuscript*
line 443. "The this" needs fixing: *sentence removed in revised manuscript*
line 645. Capitalise "smith": *amended*
line 658. Fix the author-list: *fixed*
line 678. Change Fettweiss to Fettweis: *amended*
line 714. Fix "H.-G.rensen": *fixed*
line 746. Fix "sheetL": *fixed*
Supp Info 3. Change "Assymetry" to "Asymmetry" in three places: *amended*

References

Dadic, R., P.C. Mullen, M. Schneebeli, R.E. Brandt, and S.G. Warren, 2013: Effects of bubbles, cracks, and volcanic tephra on the spectral albedo of bare ice near the Trans-Antarctic Mountains: implications for sea-glaciers on Snowball Earth. J. Geophys. Res. (Earth Surfaces), 118, doi:10.1002/jgrf.20098.

Dang, C., R.E. Brandt, and S.G. Warren, 2015: Parameterizations for narrowband and broadband albedo of pure snow, and snow containing mineral dust and black carbon. J. Geophys. Res., 120, doi:10.1002/2014JD022646.

Müller, T., A. Schladitz, A. Massling, N. Kaaden, K. Kandler, and A. Wiedensohler (2009), Spectral absorption coefficients and imaginary parts of refractive indices of Saharan dust during SAMUM-1, Tellus B, 61, 79–95.

Patterson, E. M., D. A. Gillette and B. H. Stockton (1977), Complex index of refraction between 300 and 700 nm for Saharan aerosols, J. Geophys. Res., 82, 3153-3160.

Wagner, R., T. Ajtai, K. Kandler, K. Lieke, C. Linke, T. Müller, M. Schnaiter, and M. Vragel (2012), Complex refractive indices of Saharan dust samples at visible and near UV wavelengths: a laboratory study, Atmos. Chem. Phys., 12, 2491-2512.

---

## Author Comment (AC2) · 4 Jul 2019

**Reviewer 1: Response to Review**

*We thank the reviewer for their efforts in reviewing our paper. We have reflected on the comments and addressed them as detailed below. We have made substantial changes to the manuscript. Please note that we have uploaded one revised manuscript to satisfy both sets of review comments.*

**1) The title is too general, given that the algae observations are from a single field site and the surface runoff estimates are based on a transect with forcing data from three automatic weather stations. After reading the whole manuscript, I find that it is not so convincing that the field observations of this site are representative for the entire Greenland Ice Sheet**

*With respect, we point out that nowhere in this manuscript do we claim that our field observations are representative of the whole ice sheet. Our analysis is constrained to the south western portion of the ice sheet, meaning we only assume our field observations are representative of the region delineated by Tedstone et al. (2017) with the southerly extent at 65 degrees and the northernmost extent at 70 degrees latitude, as shown in our Figure 4 (Figure 5 in revised m/s) and in our new figure 1. We drew the extent of our upscaling on Fig 1 to be clear.*

*That said, we have also visited a field site in the North West, near Upernavik, where we observed the same species of glacier algae acting as the primary albedo reducer on the ice surface. Detailed reports of algal abundance, mineralogy and albedo are not yet available, but this site was far outside of upscaling area used in this study and gives support to our upscaling within the south-western ablation zone. We agree that the scope of the upscaling could be made clearer in the manuscript, so we have altered the title and the terminology used throughout the manuscript to "south-western ablation zone" and adjusted the methods and discussion to better explain our spatial boundaries (detailed in response to later comments).*

**2) The estimation of surface runoff caused by algae over the western GrIS is not clearly presented. Did you use the classification results from UAV and Sentinel-2 to estimate the surface runoff over the entire western GrIS? How? Your sentinel-2 and UAV imagery cover a very small portion of the western GrIS. How did you generalize the results? It seems that you only modeled the surface runoff over three points along a transect where forcing data from weather stations are available, and then extrapolated the runoff estimates across the whole area based on elevations. I believe there are lots of uncertainties here, even without considering the spatial heterogeneity of surface albedo. So in your abstract where you concluded that 'algal growth led to an additional 5.5-8.0 Gt of runoff from the western sector of the GrIS in summer 2016', an uncertainty estimate is mandatory**

*We appreciate the comments and agree that this process could have been better explained in the manuscript, and that the reviewer has raised some important and interesting points. We have considered them deeply and adjusted our calculations and discussion accordingly, as discussed below. There is a lot to unpackage in this comment so we have separated our responses to each point into responses a) to d) below:*

a) *The estimation of surface runoff caused by algae over the western GrIS is not clearly presented*

*We acknowledge that the reviewer did not find this part of the manuscript clearly presented so we have endeavoured to improve the presentation style and hope that the revised version is much more clear. In response to other comments we have made significant updates to the runoff modelling process and made extensive manuscript refinements to make the runoff modelling clearer and more tractable.*

b) *Did you use the classification results from UAV and Sentinel-2 to estimate the surface runoff over the entire western GrIS? How?*

*As explained above, our analysis was restricted to an area between 65º N and 70º N. We have adjusted our runoff modelling method in response to other comments in this review, now incorporating a point-surface energy balance model to quantify melting and associated uncertainty. We think this is now clearly explained in the revised manuscript in sections 2.6 and 2.10.*

c) *Your sentinel-2 and UAV imagery cover a very small portion of the western GrIS. How did you generalize the results?*

*Our UAV imagery covers a small part of the western GrIS but our Sentinel-2 data covers a much larger area. In the revised manuscript we focus on one high quality S-2 tile covering our field site without cloud obscuring the ice surface. Adjacent tiles were cloudy on the days surrounding our UAV flight. We consider the spatial coverage of algae in this tile to be representative of the south western ice sheet margin in our latitudinal range of interest. We also point out that our latitudinal range of interest is the same as that identified by Tedstone et al (2017) and only extends North as far as 70º N, an area we consider to be well represented by our remote sensing, and we do not draw any conclusions about ice*

*outside of this area in our paper. However, we do concede that this could be made more clear in the manuscript, and we have adjusted the terminology from "western GrIS" to "south-western GrIS" throughout the manuscript.*

d) It seems that you only modeled the surface runoff over three points along a transect where forcing data from weather stations are available, and then extrapolated the runoff estimates across the whole area based on elevations. I believe there are lots of uncertainties here, even without considering the spatial heterogeneity of surface albedo. So in your abstract where you concluded that 'algal growth led to an additional 5.5 - 8.0 Gt of runoff from the western sector of the GrIS in summer 2016', an uncertainty estimate is mandatory

*The runoff modelling was carried out as described in van As et al. (2017) and in our methods section and is indeed forced by meterological data from three automatic weather stations, apart from albedo which is from MODIS and varies spatially. In van As et al.'s (2017) study they compared the performance of the model with independent observations and found the error to be negligible in the bare ice zone, with non-negligible error (underestimating runoff from snow by 0.5 m w.e.) above the snow line. Since we consider the K-transect to be broadly representative of the area over which we have upscaled, and all our runoff modelling has been limited to the bare ice zone, we maintain that this approach is valid. A much clearer quantification of uncertainty in the radiative forcing and melt attributed to algae is now included in the revised manuscript.*

**3) Cook et al. attributed the albedo reduction to glacier algae because mineral dust was considered as less effective on albedo reduction based on the radiative transfer modeling. However, the surface meltwater itself has a significant role in reducing thealbedo, which was not considered and evaluated**

*It is true that we have not directly isolated and quantified the effects of meltwater ponding on albedo. However, our empirically derived estimates of albedo reduction and radiative forcing necessarily represent the total albedo reducing effects of the glacier algae including feedbacks to the optical properties of the underlying ice and associated meltwater. We have tried to explain the tight feedbacks between algal growth and ice surface development in the manuscript and the reasons why the albedo reducing effects of physical and biological processes are intricately interwoven. To borrow terminology from a previous paper, this is an example of a "biocryomorphic" process whereby the ice and the organisms growing on it co-evolve (Cook et al. 2015) – it is difficult to determine to what extent ice melts due to the glacier algae growing on it or the glacier algae grow due to nutrients and liquid water liberated from ie that is already melting, but we consider it safe to assume it is a combination of both, which is what makes algal growth an especially powerful albedo reducer, but also makes quantification of the albedo reducing effects of a single element in the system challenging. That said, we have provided the BioSNICAR_GO model that employs geometrical optics in preference to Mie scattering to simulate the optical properties of the ice and the algae. In theory, increasing the ice grain size could adequately simulate the accumulation of interstitial meltwater since the refractive indices of ice and water are very similar and interstitial water replaces air-ice interfaces with water-ice interfaces. However, this would necessarily assume that meltwater accumulates in situ because we do not have a good method for simulating percolation and throughflow of meltwater, but we know it will vary dramatically according to the porosity of the weathered layer. Therefore, we consider the uncertainties in accounting for meltwater directly to be too large to include it in a quantitative manner in this study, but we point out that we have accounted for the "total" glacier algae albedo reducing effect that includes physical, biological and hydrological processes locked in a tight feedback, and explained these feedbacks in the manuscript. We have added more discussion of these concepts to the revised manuscript.*

**4) Many critical details on methods and results are missing, which need clarifications.
See specific comments below.**

*We address these individually below.*

**5) The overall writing needs to be improved, particularly the writing style. The methodand result parts are poorly structured, which seem like a simple but loose stackingof various materials, while the logical linkages between different parts are weak andnot clear. This study involves several different components, including fieldwork, radiative transfer modeling, image classification from UAV and Sentinel-2 data, and surfacerunoff modeling using 'an SMB model.' In the second section 'Field sites and meth-ods,' all the materials related to those components are just put together, which are verydifficult for readers to follow. There are also many redundant descriptions betweenthe method and result parts, which seems like that this manuscript was not thoroughly proofread. There are also some grammar errors and typos**

*We acknowledge the reviewer's comments on the writing style and have endeavoured to improve the manuscript for clarity. We hope that the revised manuscript is easier to follow.*

**6) The most recent literature about ice algae mapping using remote sensing data is not cited and discussed, like:Wang, S., Tedesco, M., Xu, M., & Alexander, P. M. (2018). Mapping Ice Algal Bloomsin Southwest**

**Greenland From Space. Geophysical Research Letters, 45(21), 11,779–11,788. Huovinen, P., Ramírez, J., & Gómez, I. (2018). Remote sensing of albedo-reducing snow algae and impurities in the Maritime Antarctica. ISPRS Journal of Photogrammetry and Remote Sensing: Official Publication of the International Society for Photogrammetry and Remote Sensing, 146, 507–517.Both these two papers are using remote sensing data to detect snow/ice algae. Although the second paper is dealing with snow algae, the first paper is utilizing the chlorophyll-a signature to map ice algae from satellite imagery over the southwestern GrIS. Your presented field data and radiative transfer modeling results (particularly your Figure 2A) are consistent with Wang et al. (2018) who used the reflectance ratio between 709 and 673 nm to quantify the ice algae**

*The reviewer is correct that we had not sufficiently cited those papers, for the following reasons:*

*1) Huovinen et al. 2011 focuses entirely on snow algae. This is a drastically different system to the glacier algae we concentrate on in our study, and we considered discussion of this paper to be tangential. However, we have incorporated a brief mention of the paper, and why their methods cannot be transferred from snow to ice ecosystem in the introduction.*

*2) Wang et al. (2018) used the vegetation red-edge to map glacier algae; however, the vegetation red-edge can be vulnerable to false positives due to mineral dusts, especially where the proportion of red minerals is high (Cook et al 2017). We attempted to use the vegetation red-edge on our spectra and achieved only ~80% accuracy for identifiying algal ice compared to our random forest classifier that achieves >95% accuracy. Furthermore, the red-edge can only provide a binary classification where a pixel is either biological or not. Our classifier can map a range of ice surfaces – we chose five classes that we consider to be representative of the majority of the ablation zone. Wang et al. (2018) used Sentinel-3 OLCI data that has a ground resolution of 300 m to quantify algae. At this ground resolution the uncertainty due to spatial heterogeneity is surely large, especially because the underlying ice optics and mixing of various impurities were not accounted for, and there is no direct ground validation that could ameliorate some of these various sources of uncertainty. For these reasons we did not wish to benchmark our work against that paper.*

*However, we have now added discussion of both papers to our revised manuscript.*

**Specific comments.**

**Introduction: This section should be expanded, at least including a more detailed literature review about the current research progress and efforts on ice algae and their relationship with albedo and surface melting.**

*We have expanded the introduction as requested*

**Line 62. The study by Wang et al. (2018) should be cited, which used the spectral signature of chlorophyll-a to map ice algae in Greenland.**

*We have added the citation*

**Line 78. Adjust your figures. It's a bit odd to put your first figure in the text as Figure 3c.**

*We have adjusted our figures*

**Line 86. Do you have multiple sites? What's your sampling area size? Did you take point measurements at different places within a specific area? Explain those details in your field site description part.**

*We have updated our methodology to clarify these points*

**Line 99. Update your reference 'Cook et al. (2017b)' in the reference list.**

*The reference has been updated*

**2.3 Biological Measurements. Did you also differentiate different species when counting the cells?**

*Yes we differentiated between the species, this is now made explicit in the revised manuscript*

**Line 126. What is PSD? Particle size distribution? Do not use abbreviation when you use the term first time in your manuscript.**

*PSD stands for particle size distribution. The manuscript has been amended so that the abbreviation is defined on first use.*

**Line 135. Provide more details on how you used the ASD to measure the surface reflectance of the materials pressed on the microscope slide, such as your measurement setup, the field view of the ASD probe, and the background material (white, grey, or black) where you put your microscope slide.**

*The bare fibre of the ASD field Spec was inserted into the viewing port of the integrating sphere. The integrating sphere is a hollow sphere with the inner surface covered in a near-Lambertian coating that makes the light reaching the fibre diffuse. The sphere has a set of circular apertures that can be opened to insert samples and to allow a light source to illuminate the interior. One aperture was opened and the sample was pressed into it. This aperture was positioned at the bottom of the sphere so that the aparatus could be lowered onto the sample which was arranged into an optically thick layer (~3mm thick) on a microscope slide which in turn was on an opaque white stopper beneath, meaning the sample could remain in a stable position throughout the measurement.*

*This is now clarified in the revised manuscript.*

**Line 144. Any references for the BioSNICAR_GO? Is BioSNICAR developed for this study? Provide more details.**

*BioSNICAR_GO was developed for this study, building major advances on top of the BioSNICAR software published in Cook et al. (2017b). We have added more details to the revised manuscript including a new figure showing a schematic of the model structure. We direct the reviewer to the documentation provided in our repository at the doi provided or on Github at [www.github.com/jmcook1186/BioSNICAR_GO](www.github.com/jmcook1186/BioSNICAR_GO)*

**Section 2.5 paragraph 2. As I asked before if you considered the different cell numbers of different algal species, did you take into account the different shapes of the Ancylonema nordenskiöldii, Mesotaenium berggrenii? Ancylonema nordenskiöldii is filamentous while Mesotaenium berggrenii is unicellular. If you consider the geometrical optics, how would these two different shapes affect the radiative transfer modeling? Can you comment on the sensitivity of radiative transfer modeling on algal cell shapes? Can you combine sections 2.5 and 2.6?**

*We did include different algae in our radiative transfer model to account for both of the dominant species. We found the albedo to be insensitive to cell dimensions within a realistic range of lengths and widths. We simulated chains of cells as continuous cylinders as mentioned in the existing text, supported by Hillebrand et al (1999) and Lee and Pilon (2003). We have added the following explanatory text to our manuscript to clarify:*

*"This was undertaken for a range of cell dimensions that are now available in the lookup library for BioSNICAR_GO. For this study, we included two classes of glacier algae representing Mesotenium bergrenii and Ancylonema nordenskioldii with length and width and also the relative abundance of each species matching the means measured in our microscopy described in Section 2.3. In simulations not shown here the albedo was relatively insensitive to the dimensions of the cells within a realistic range of lengths and widths. For example, in a simulation with constant ice optics (snow of constant grain size 400 µm, density 400 kg m$^{-3}$ and snowpack thickness 50 mm) and a biomass mixing ratio of 10$^5$ ppb changing the length of the algal cells from 10 to 40 um changed the albedo by less than 0.4%. The albedo was more sensitive to cell width; with the same ice optics and biomass mixing ratio as described above and an algal cell 40 µm long, changing the cell width from 4 µm to 12 µm changed the surface albedo by 0.8%. This low sensitivity is likely because all of the cells considered here are large from a radiative transfer perspective."*

**Line 187. 'utilise 5% of this ...' any references for this?**

*In the absence of in situ measurements of photosynthetic efficiency in our study or any past study, and acknowledging high sensivity to local environmental conditions, we took a realistic value from the literature for other photosynthetic microalgae. This was also corroborated by personal communications with Chris Williamson (Univ. Bristol) who is preparing a manuscript on the photosynthetic efficiency of these specific algae. We chose to take a value at the upper end of the realistic range because this translates into a conservative estimate for algal melt acceleration (since a larger portion of the energy is used photosynthetically rather than transferring into the surrounding ice. We have added further details to the revised manuscript.*

**Line 191. determine -> determined**

*corrected*

**Section 2.7 paragraph 2. This paragraph is overall difficult to read. Could you use some equations to show your calculations?**

*This section has been rewritten for clarity.*

**2.8 UAV remote sensing. When did you conduct the UAV mapping, the same time with your field spectral collection? Also specify the multispectral camera parameters, like band wavelength, bandwidth and so on.**

*We have added further details regarding the camera specification.*

**Line 210. oin -> in**

*corrected*

**Line 216. What do you mean by time-dependent regression?**

*This is now explained in the manuscript as follows:*

*"We then converted from radiance to reflectance using time-dependent regression between images of the MicaSense Calibrated Reflectance Panel acquired before and after each flight (i.e. a regression line was computed between the reflectance of the white reference panel at the start and end of the flight and used to quantify the change in irradiance during the flight).This was used to calibrate the UAV multispectral images to reflectance."*

**Line 222. What do you mean by 'generally good'? Move this to the results or discussion part and make clarifications.**

*There was generally close agreement between the albedo estimates although in some cases variations in the radiometric calibration of each sensor led to some non-negligible differences. This is fully discussed in a separate manuscript recently submitted to this journal.*

**Section 2.9. It's very odd to have just two sentences in a single section. You should combine this with classification, or introduce more about the Sentinel-2 data. How does the Sen2Cor perform? Can you provide a figure showing the atmospherically corrected surface reflectances of Sentinel-2 data? What are the acquisition dates of your Sentinel-2 data?**

*These details have now been added to the revised manuscript*

**Line 231. remove the word 'novel' as random forest classification has been widely used.**

*Whilst this is true, we maintain that the application of the random forest classifier to mapping glacier algae, and the idea of training on reduced field spectroscopy data to overcome issues of spatial heterogeneity, are highly novel.*

**Line 234. What tiles? Sentinel-2?**

*Yes, this is now explicit in the manuscript*

**Line 236. How did you reduce the ASD spectra to the UAV bands considering the difference between their bandwidth? Please clarify.**

*We used the ASD wavelengths equal to the center wavelength of the camera bands.*

**Line 240. This part is not clear. The reflectances at five wavelengths (reduced from ASD spectra?) were used as the feature vector, what's your classification vector? How many classes and what classes you were training? How many training samples do you have?**

*This information is all available in the manuscript and in the documented code in our repository, but we apologise if it is not sufficiently clear. To clarify, we trained our model on ASD reflectance reduced to either 5 wavelengths (in the case of UAV imagery) or nine wavelengths (in the case of Sentinel-2 imagery). The reflectance at each of these wavelengths was the feature vector, and the surface class associated with the sample surface was used as the classification vector. There are six possible surface classes in our model: HA, LA, CI, CC, WAT, SN referring to heavy algal bloom, light algal bloom, clean ice, cryoconite, water and snow. We had 174 training examples concentrated into HA, LA, CI and SN, with fewer examples of WAT and CC. WAT and CC are relatively easy for the classifier to identify,*

*being very low albedo with flattish spectra. We ensured that there were mutliple examples of each class in both the training and test sets. The diagram attached here as Figure 1 has been added to our supplementary information.*

*We consider this method of training on ground spectra to be a significant advance from prior classification techniques in the cryosphere literature for several reasons:*

*1) For every sample, we have complete confidence in the labelling because we have removed the surface ice and analysed it in the laboratory as well as recoridng on-site metadata*

*2) We have minimised error due to surface heterogeneity. Since the surface classes are patchy, often with relatively small length scales, post-hoc labelling of aerial imagery is more likely to either be misclassified or classify ambiguous spectra. Our method ensures that the reflectance spectra is derived from a homogenous surface with a definite label as explained in point 1).*

*3) Training on reduced hyperspectral data makes the whole classification method sensor-agnostic, i.e. the training set could easily be resampled to train for a different satellite platform or multispectral camera, greatly enhancing the reuseability of the hard-won ground data.*

*4) We have provided a completely open code resource where the spectral library can (and will) be appended to as more spectra (gathered using any of the industry standard VIS-NIR spectroradiometers such as the ASD Field Spec) become available from our follow-on projects and – we hope - other researchers. The model can then be retrained sequentially as the training set grows. That said, although our existing training set is small, the performance in hold-out training is very good.*

**Line 269. clarify the 'rolling-window approach' or use reference**

*The rolling window approach is explained in full in Tedstone et al (2017) as follows:*

*"Each year, we identified the first rolling window at each pixel that contained atleast 3 days of bare or dark ice (not necessarily consecutive) and 0 days of non-bare or non-dark ice, which therefore per-mitted up to 4 days of cloud cover in the window. We then selected the first day of bare or dark ice appearance fromwithin the chosen window. This windowing strategy enabled us to minimise the likelihood of false-positive identificationof bare and dark ice appearance dates which would have oc-curred if only looking at daily observations in isolation andalso allowed us to ameliorate for cloud cover."*

*We have added citation to this paper in the text.*

**Line 271-276. Rewrite this part. Specify your surface class.**

*We have rewritten this section for clarity*

**Line 283. 'surface albedo is adjusted from MODIS...' How?**

*"adjusted" is a mistype – removed. The albedo is simply the albedo predicted by MODIS MOD10A1.*

**2.12 Runoff modeling. What's the relationship between your remote sensing classification and runoff modeling. Did you use the classification results to constrain your runoff modeling? Uncertainty estimation should be included. The surface runoff estimation is not rigorous considering the albedo difference.**

*We have added significantly to the methodology and discussion to address these points in the revised manuscript.*

**Line 301-307. Combine these texts with your field site and measurements description in section 2.**

*We have adjusted the manuscript accordingly.*

**Line 312-313: Does the number after '+/-' mean standard deviation? Change the symbol to '±'. What standards did you take to separate the samples into those four classes? Can you show the histogram of your samples and the separating boundaries of the classes?**

*The surface classes were separated by visual assessment of the ice surface made at the same time as the albedo measurements were made and the sample removed. The stated mean +/- standard deviation for the cell abundance for*

*each surface class was derived from counting cells in microscope images of melted ice samples for those same surfaces, as described in the methodology. To demonstrate the separation between the surface classes I have provided a histogram as Figure 2 attached here, please note the unequal bin widths as the cell counts cover 4 orders of magnitude.*

**Line 360. Plot the absorption spectrum for the purpurogallin pigment, with other photosynthetic and photoprotective pigments.**

*We have added a plot showing the absorption spectra for the algal pigments to Supplementary Information*

**Line 374. Be cautious about making this conclusion based on your field measurements over just one specific area.**

*We have made major revisions to the manuscript that address this comment*

**Line 383-385. You didn't take any biological measurements over those 'wavy' areas. Generalizing your single-site observation to the entire GrIS is inappropriate.**

*The wavy areas are common to the Dark Zone and are presented by Wientjes et al (2010, 2011) as evidence in favour of geological darkening. We simply provide counter evidence that suggests that while the wavy pattern may well signify geologic controls on spatial patterns within the Dark Zone, we do not believe the darkening to result from the presence of dark minerals – more likely the minerals stimulate algal growth*

**Section 3.5. This part needs to be rewritten. Many texts should go to the methods section. I find that this manuscript has a lot of those redundant descriptions. Some texts should be in the previous section but were put in the results section. Can you plot out the spectra (reduced to UAV and Sentinel-2 bands) of the four different classes, in comparison with your original ASD spectra? The ASD spectra may well differentiate four different classes, but the reduced spectra would mask out lots of unique spectral
signatures considering the coarse spectral resolutions. Otherwise, your classification (no matter what advanced methods to be used) is not supported. Besides, you should also plot out the real UAV and Sentinel-2 spectra, in comparison with your ASD spectra.**

*We have considered this and taken into account the important difference between hyperspectral data generated using the ASD Field Spec and the multispectral data acquired by our camera or Sentinel-2, and we do feel this is explained in the manuscript. With respect we point to the following passage that clearly states this for the UAV:*

*Line 235: "Our directional reflectance measurements were first reduced to reflectance values at five key wavelengths coincident with the centre wavelengths measured by the MicaSense Red-Edge camera mounted to our UAV (blue: 0.475, green: 0.560, red: 0.668, red-edge: 0.717, NIR: 0.840 μm)."*

*and then for Sentinel-2:*

*Line 249: "This protocol was then repeated for Sentinel-2 imagery. In that case the directional reflectance data was reduced to eight bands coincident with the centre wavelengths measured by Sentinel-2 at 20m ground resolution (0.48, 0.56, 0.665, 0.705, 0.740, 0.788, 0.865, 1.610 μm)."*

*However, we have rewritten for clarity as explained in the response to a previous similar comment above. The trained classifier is not picking out spectral features only present in hyperspectral data. We are sure of this because the classifier is trained on the reflectance at only the reduced set of wavelengths.*

**Line 444. The this -> The.**

*Thank you, amended.*

**Section 3.8. Discuss the potential impact of meltwater itself.**

*We have added discussion of this to the revised manuscript*

**Table 1. explain the abbreviation in your table title or put a note on this.**

*Amended*

**References:**

Cook J, Edwards A and Hubbard A (2015) Biocryomorphology: Integrating Microbial Processes with Ice Surface Hydrology, Topography, and Roughness. *Front. Earth Sci*. 3:78. doi: 10.3389/feart.2015.00078

Cook JM, Hodson AJ, Gardner AS, Flanner M, Tedstone AJ, Williamson C, Irvine-Fynn TD, Nilsson J, Bryant R, Tranter M. 2017. Quantifying bioalbedo: A new physically-based model and critique of empirical methods for characterizing biological influence on ice and snow albedo. The Cryosphere: 1–29. DOI: 10.5194/tc-2017-73. 2017b

Hillebrand H, Dürselen C-D, Kirschtel D. et al (1999) Biovolume calculation for pelagic and benthic microalgae. J Phycol 1999;35:403–24

Lee, E., Pilon, L. (2013) Absorption and scattering by long and randomly oriented linear chains of spheres. Journal of the Optical Society of America, 30 (9): 1892- 1900

Tedstone, A.J., Bamber, J.L., Cook, J.M., Williamson, C.J., Fettweis, X., Hodson, A.J., Tranter, M., 2017, Dark ice dynamics of the south-west Greenland Ice Sheet, The Cryosphere Discuss, doi: https://www.the-cryosphere-discuss.net/tc-2017-79/

Wang, S., Tedesco, M., Xu, M., & Alexander, P. M. ( 2018). Mapping ice algal blooms in southwest Greenland from space. *Geophysical Research Letters*, 45, 11,779– 11,788. https://doi.org/10.1029/2018GL080455

Wientjes, I. G. M., Oerlemans, J. (2010) An explanation for the dark region in the western melt zone of the Greenland ice sheet, The Cryosphere, 4, 261–268, https://doi.org/10.5194/tc-4-261-2010.

Wientjes, I.G.M., Van de Wal, R.S.W., Reichart, G.J., Sluijs, A., Oerlemans, J. (2011). Dust from the dark region in the western ablation zone of the Greenland Ice Sheet. The Cryosphere, 5, 589–601, 2011, doi:10.5194/tc-5-589-2011

---

## Author Comment (AC3) · 4 Jul 2019

[revised manuscript text omitted]

---

## Author Comment (AC4) · 6 Jul 2019

The Cryosphere

**Cook et al. (2019): Glacier Algae accelerate melt rates on the southwestern Greenland Ice Sheet**

**Revised Supplementary Information**

**S1: Schematic diagram of the BioSNICAR\_GO model structure**

**Supp Info 2: Mineral dust sampling and particle size distribution (PSD).**

High algal biomass ice samples were collected in sterile sample bags and melted at ambient temperatures (5-10  $\square$ C). The thawed samples were filtered onto glass fiber filters (0.7  $\hbar$ m pore size), from which the solids were removed into a glass jar using a stainless steel spatula. In 50 mL centrifuge tubes, the samples were treated using 30% H2O2 (w/w) (Honeywell FlukaTM) to remove the organic fraction. The samples (1-2 g) were sonicated (VWR ultrasonic cleaner) in 45 mL of the H2O2 treatment for 10 min to disaggregate the material. The samples were left in the H2O2 treatment for 48 h, after which they were centrifuged for 10 min at 4000 rpm (Eppendorf centrifuge 5810). The supernatant was removed, and the H2O2 solution was replaced. This process was repeated up to ten times until no more organic oxidation was observed. The remaining mineral fraction was washed three times in water (Sartorius ariumII) pro ultrapure water), with centrifugation after each wash.

A 5 mg of H2O2-treated sample was suspended in 10 mL of ultrapure water. The sample was sonicated to disaggregate the grains. The suspension was dispersed onto a 0.2  $\hbar$ m polycarbonate filter (Sartorius Track-Etch Membrane, 0.2  $\hbar$ m). Once dry, a section of each filter was adhered to a stainless steel SEM stub using an adhesive carbon tab. The sample was coated with 8 nm of Ir (Agar high resolution sputter coater). The PSD was determined using a Zeiss Ultra Plus field emission scanning electron microscope (FE-SEM) operated at 20 kV. Automated particle counting software was used to determine the PSD in an area of approximately 1 mm2.

---

## Author Response (AR2)

**re: Cook et al. 2019: Glacier algae accelerate melt rates on the south-western Greenland Ice Sheet Authors response to reviewer comments, round 2.**

Dear editor and reviewers,

we are very grateful for the second set of reviews of our paper. We have endeavoured to address all the comments made by both reviewers to the best of our ability. Most notably, we have removed the DISORT inversion from our study entirely, as Reviewer 1 pointed out strange behaviour at the short visible wavelengths that we could not resolve. This has been replaced by an alternative method that uses measured particle size distributions and mineral mass-fractions from field data along with mineral refractive indices from the past literature to estimate optical properties from the local dusts.

In this document, we provide point-by-point responses to the reviewer comments and a copy of the manuscript with the main changes highlighted. We have also submitted a revised manuscript and substantial new supplementary information as well as updating our data and code repositories (we highlight to the editor that we have held back from minting doi's for these repositories until the end of the peer-review process so that we can update according to reviewer queries).

We sincerely thank the reviewers for their time and effort in reviewing this paper. We consider the paper to be much better as a result and we hope that this manuscript satisfies the concerns raised in the comments.

Kind regards,

Joseph Cook and co-authors

**Second review of Cook et al. for TCD, August 2019, by Stephen Warren "Glacier algae accelerate melt rates on the south-western Greenland Ice Sheet"**

**Recommendation: Major revision required.**

This paper is still not ready for publication. Concerning the melting ice surface of the West- Greenland ablation zone in summer, the authors argue that addition of dust causes the ice albedo to increase, so that any reduction of ice albedo by light-absorbing impurities would be due to algae. This denial of a role for dust in reducing albedo is based on several questionable arguments, which I will point out in this review. Most of these questionable arguments can be classified into one of three classes of disconnects: (a) The authors' response to my review, saying how the paper was changed, does not correspond with what the revised paper actually shows. (b) The way figures are described in the text does not correspond to what the figures actually show. (c) References cited in support of a claim do not actually provide the claimed evidence.

I do not doubt that algae absorb sunlight, but I do doubt the quantitative attribution of albedo change on Greenland to algal abundance. A lot of work went into this project, and I would like to see it come to more robust conclusions. It has the potential of becoming an important paper.

- Thank you for another round of review comments. In this document I will address each comment point by point. We have re-examined our paper in detail and overall we agree about the DISORT inversion issues. Thanks to the comments we ran a series of tests and found some unexpected behaviour in the model and have been unable to produce model outputs that we have sufficient confidence in to justify the model's inclusion in this resubmission. For this reason, we have removed the DISORT inversion from the study. Instead, we have generated optical properties for three bulk mixtures of dusts that approximate the mineralogy and size distribution measured in the field and incorporated them into our model. In addition we have included a sensitivity study as supplementary information (Supp Info 8) that includes previously-published mineral dusts that have different mineralogies and size distributions. More details and responses to individual comments are provided below.

**Major comments**

(1) The major comment of my first review was to point out that the imaginary part of the dust refractive index should decrease across the visible spectrum from 300 to 700 nm, not increase as shown in the first submission. In their Response to Reviewers, the authors now thank me for "pointing out the unrealistic refractive index", but then in their revised manuscript their dust imaginary index still increases across the visible (by a factor of 2.7 from 300 to 700 nm), as shown in the new Figure 3C. [An example of Disconnect Type "a".]

- We appreciate the reviewer's concerns regarding the DISORT inversions. We have reviewed the modelling in detail and we agree that the retrieval was not correct, and we have not been able to generate an imaginary refractive index that we have complete confidence in reporting in the revised manuscript. The mineral dusts did not look blue to the eye - they were very fine, near-white, dominated by quartz and feldspar minerals with very low abundance of red minerals. This is confirmed by mineralogical analysis undertaken by McCutcheon et al. (in preparation) and is consistent with previous literature (e.g. Wientjes et al. 2011). For these reasons, we have decided to remove the DISORT inversion from our paper and instead we have generated "synthetic" bulk refractive indices from our measured particle size distribution, measured relative mineral abundances and imaginary refrative indices for those minerals gathered from past literature. We have also persisted with our sensitivity study that includes typical Saharan dusts from SNICAR and low, medium and high-hematite dusts from Polashenski et al (2015), although we have moved this into Supplementary Information 8 to keep the main message in the manuscript clear. We consider this to be the most robust support for our conclusions reaarding the albedo-lowering effects of mineral dusts and alage that we can feasibly produce with the available data. As you will see in the revised manuscript, this new methodology supports our original conclusion that mineral dusts have a very small direct albedo-reducing effect on the south-western Greenland Ice Sheet.

The relevant reworked sections of the manuscript are sections 2.5, 2.6 and 3.2, Table 1, Table 2, Fig 3B,C and Supp Info 8.

(2) I don't want this extraordinary claim of blue dust to enter the literature on the composition of Greenland ice without further evidence. What mineral composition gives it the blue color indicated by Figure 3C? In their response to my review, the authors indicate that the mineralogy of the dust is consistent with Figure 3C, and that a paper on this topic is soon to be submitted by one of the authors (McCutcheon). A brief summary of the mineralogy in that forthcoming paper should be included in this paper; it could be cited as "unpublished data" or "manuscript in preparation".

- We fully appreciate and agree. As explained above, we have eliminated the DISORT inversion that was generating this unusual refractive index. New data from McCutcheon et al. (in preparation) is now used to generate a "synthetic" bulk refractive index for our local mineral dusts and that paper is cited as "in preparation" as suggested. The mineralogy of these simulated local dusts is presented in the new Table 1. This is detailed in sections 2.5, 2.6 and 3.2.

(3) The authors have ignored the request in my Major Comment #1, in which I asked the authors to show the computed albedo effect of the measured dust concentration; they continue to show just the computed albedo effect of arbitrary amounts of dust (100, 300, 500 ppm), and similarly for arbitrary concentrations of algae. At least a table is needed, giving measured dust and algal concentrations (ppm by mass). The numbers of cells are shown in Figure 2C, but these need to be combined with algal-cell size distributions to get the mass.

- We have now incorporated new dust concentration data from McCutcheon et al (in preparation) into our study to address this comment – specifically the mean (342  $\mu g_{dust}/g_{ice}$ ) and maximum (519  $\mu g_{dust}/g_{ice}$ ) measured dust concentrations from  $H_{bio}$  sites. We also estimated the algal cell mass-mixing ratio from our microscope images. The mean cell abundance was 2.9 x 104 cells/mL and the maximum was 4.91 x 104 cells/mL. We assumed a cell density of 0.87 g cm-3 (Hu, 2014) and an ice density of 0.917 g cm-3 to calculate the mean and maximum mass mixing ratios of 349 and 646  $\mu g_{algae}/g_{ice}$ .

We have also continued to include additional hypothetical mass mixing ratios in our sensitivity study in Supp Info 6 to demonstrate that our conclusions are robust to a range of mass-mixing ratios.

(4) Lines 272-273. Tedesco et al. (2013) is cited as indicating "a lack of red mineral phases". In fact, Tedesco's Figure 6a shows that both dust and algae are "red", and that dust is redder than algae; Tedesco speculated that the goethite they found in their samples had dehydrated to hematite in the drying and heating process. Goethite is the hydrated form of hematite; it is not as absorptive as hematite but its absorption coefficient likewise decreases across the visible. It is true that goethite has a yellow appearance rather than red, but it is misleading to cite Tedesco as finding "a lack of red mineral phases", since the present authors are using the adjective "red" here to characterize the spectral slope of reflectance, which increases toward the red for both goethite and hematite. [Disconnect type "c"]

- We respectfully disagree with the reviewer on this point, for the following reasons:

1) Tedesco et al. (2013) identified an average of only 0.3 % goethite in the Greenland cryoconite samples, which, even if present as hematite prior to transformation during sample processing, would likely not be sufficient to account for the "red" colour in question.

2) Figure 6a in Tedesco et al. (2013) does not show that Greenland surface dust is redder than algae. The samples studied in Tedesco et al. (2013) were from cryoconite rather than surface ice, and therefore are unlikely to have contained glacier algae in abundances comparable to those we have measured in dark surface ice. In fact, the organic matter contained in the cryoconite is largely bacterial, cyanobacterial, humic substances, extracellular polymers and necromass – optically very different from surface algae.

3) Furthermore, the "red" colour measured in the mineral dust by Tedesco et al. (2013) is not representative of the natural material due to the manner in which the samples were processed. The samples were heated to 500 and 1000 degrees C, which is entering hornfelsic grade metamorphic conditions. This will have altered Fe-containing hornblende and pyroxene mineral phases thereby generating the reported "red" colour, which cannot be used to draw conclusions about the true reflectivity of the mineral dust in situ. In contrast, we

used a suitable chemical treatment rather than heat to remove organic matter from the samples and have data likely to represent more realistic in situ mineral optical properties.

**We have added text to the revised manuscript to make these points clear (line 556 - 564).**

(5) Lines 518-521. "The imaginary refractive index of the mineral dust sample (Fig 3C) . . . indicating . . . scarcity of red minerals in the bare ice." This comment is not forthright; there is no mention of the factor-of-2.7 increase of imaginary index from 300 to 700 nm, which indicates not merely a scarcity of red minerals but actually a dominance of blue minerals. Don't be so timid! You must highlight this strange imaginary index, and point out how it contradicts the behavior reported by Tedesco et al. 2013. [Disconnect Type "b"]

**We have reworked this section after removing the inverse modelling from our study.**

(6) Lines 539-541, discussing Figure 3B. Albedo spectra for algae "downsloping with increasing wavelength between 0.35 and 0.45 microns . . . and a gentle increase to 0.70 microns. These spectral features are consistent with our field spectra for algal ice" This is not true. The field spectrum for algal ice (Figure 2B) starts at 0.40 not 0.35, and shows albedo increasing, not downsloping, from 0.40 to 0.45 microns. And Figure 2B (field) shows a steep increase from 0.6 to 0.7 microns, whereas the dashed line in Figure 3B (model) is flat from 0.6 to 0.7. [Disconnect Type "b"]

**We have reworked this section and made more accurate descriptions of the spectral albedo.**

(7) Lines 574-575. ". . . algal cells had a greater albedo-reducing effect than mineral dusts in north-west Greenland (Aoki et al. 2013)." This summary of the Aoki paper is misleading. Aoki et al. did conclude that the imaginary index of algae was larger than that of mineral dust, but did not conclude that most of the albedo reduction was due to algae. Their total impurity mass in the ice was 1127 ppm, of which 29 ppm (2.6%) was organic carbon (algae), so ~1100 ppm was dust. Their Figure 4b shows that they could explain most of the albedo reduction by 1000 ppm dust; the remainder (which looks like about 5% to me) is then attributed to algae. [Disconnect Type "c"]

We revisited Aoki et al. (2013) and have removed the citation from our manuscript for the following reasons. There is conflation between surface impurities, cryoconite and surface dust. Their figure 2B shows what those authors consider to be a "cryoconite" surface, which is what we would now consider to be a mixture of surface dust and glacier-algal biomass similar to that included in our study. Cryoconite is properly defined as: discrete granules of biological and mineral material that typically have a very dark brown-black colour and reside on the floor of cryoconite holes unless the local energy balance conditons favour them being evacuated onto the bare ice surface. For this evacuation to happen, turbulent heat fluxes must exceed radiant heat fluxes for a period of several days (i.e. persistent cloudy conditions) so that the weathered surface ablates downwards to expose smooth solid ice and the cryoconite holes that occupy the weathered crust "melt out". The morphology of the ice shown in their Figure 2B is not consistent with this process having occurred, and this is further confirmed by the sentence "there were also cryoconite holes (waterfilled cylindrical melt-holes with cryoconite on the bottom)". If there are cryoconite holes present, any dispersed cryoconite must be spatially discrete, and the distributed impurities they refer to as cryoconite must actually be a dust and algae mixture comparable to that observed at our field site. Furthermore, they have not generated optical properties for the actual dust found at their field site, but simply imported dust optical properties from a pre-existing library where the mineralogy is assumed to be mostly illite, calcite, feldspar and chlorite derived from Asian dust samples (see Aoki et al, 2005) that are very different to our local mineralogy (and perhaps to the true mineralogy at their field site too). For these reasons, it is not helpful to compare Aoki et al.'s (2013) interpretation of their albedo data – certainly not their separation of mineral and biological effects - to ours.

Another important point is that where Aoki et al. (2013) refer to algae, they are making reference to the bright red snow algae that grows on melting snowpacks. This is taxonomically, morphologically and optically very different to our glacier algae.

Aoki et al. (2005): Sensitivity Experiments of Direct Radiative Forcing Caused by MineralDust Simulated with a Chemical Transport Model Journal of the Meteorological Society of Japan, Vol. 83A, pp. 315--331, 2005315

(8) Lines 544-546, and Figure 2C. The authors point out that mineral dust particles can "act as substrates for the formation of low-albedo microbial-mineral aggregates". This suggests an alternative explanation of the correlation shown in Figure 2C: The algae may be concentrated in patches of ice that have high mineral content, so the cell count then would be correlated with the albedo reduction caused by dust.

**We agree, this is a central question that we have attempted to address using our radiative transfer modelling.**

(9) Section 3.3, lines 577-610. This section analyzes the albedo trough centered at 1.02 microns, following Nolin and Dozier (2000). Nolin and Dozier found the 1.02-micron trough to be deeper for lower albedo (coarse-grained snow), whereas Supp Info 5B here shows the opposite, namely deeper trough for higher albedo. The 1.02-micron feature is therefore not useful for discussing "indirect effects of algae". The entire section 3.3 should therefore be shortened to just the last four lines 607-610, making a single sentence starting "Algal growth is stimulated . . . " [Disconnect Type "c"]

Yes, the relationship does show a deeper trough for higher albedo. We consider this to be an artefact of an overall lower albedo across the entire spectrum effectively "dampening" all of the reflectance features. However, the point we were making was simply that there is albedo reduction in the NIR associated with increased LAP loading as well as in the visible wavelengths, indicative of secondary or "indirect" albedo reducing processes related to grain size and shape, melt water accumulation, etc. We have dramatically shortened this section as suggested, but kept two additional introductory sentences to make the point of the paragraph clear to a wide readership, and hope that this is acceptable.

2(10). Figure 4ABC. Half of the solar energy is at wavelengths <0.7 microns, and 80-90% (depending on cloud thickness) is at wavelengths <1.0 microns. In these figures, the peculiar wiggles in the visible region, the most energetically important part of the spectrum, are squeezed into a tiny region on the far left of the figures. These wiggles need to be discussed and explained in the text, and the figure should be redrawn, for a domain 0.3-1.5 microns instead of 0-5 microns.

These figures related to our DISORT inversion and have therefore been discarded.

**Minor comments.**

line 206. "... they are large, far outside the domain of Mie scattering". Mie theory is not restricted to small size-parameters. Admittedly Mie calculations do become expensive for large size-parameters.

We appreciate that technically Mie calculations can be solved for any size of particle, but we also acknowledge that there is a huge computational cost associated with solving them for particles in the size range of our glacier algae. The Mie scattering domain is commonly described in the literature as 2 > size parameter

0.6

0.8

1.0

Figure 1: UAV reflectance plotted against ASD reflectance for each UAV band, in uncorrected (blue) and corrected (red) form.

---

## Author Response (AR3)

Joseph Cook
IBERS, Aberystwyth University
Aberystwyth, UK, SY23 3DA
joc102@aber.ac.uk
5/12/2019

**re. manuscript acceptance: Cook et al. (2019) Glacier algae accelerate the melting of the south-western Greenland Ice Sheet**

Dear Editor,

thank you for accepting our manuscript for publication. I am pleased to make the amends you requested and hope that we can now finalise this without further delays. Please see the responses to your comments below:

**1) " I would still like to see a comparison between ASD, UAV and remote sensing data in the manuscript"**

We added three figures to Supplementary Info 6 to show the inter-sensor comparisons and have now amended the manuscript as follows to make it completely clear:

L387: "Comparisons between the directional reflectance spectra gathered using the ASD field spectrometer and those measured using the UAV and Sentinel-2 are provided in Supp Info 6."

**2) "… how the ASD data has been reduced to the discrete bands? Was the spectral response of the sensors considered?"**

This was already explained in the manuscript but we now explain this explicitly as follows:

L391: "Directional reflectance data gathered using the ASD field spectrometer was reduced to only those nine wavelengths coincident with the centre wavelengths measured by Sentinel-2 at 20m ground resolution (0.480, 0.560, 0.665, 0.705, 0.740, 0.788, 0.865, 1.610 2.190 μm)."

**3) "Other details"**

We have provided the totality of our underlying datasets, codes, trained models, metadata, images and methodological details in our repository. We have put considerable effort into curating this to make it easily navigable by other users. We have now minted digital object identifiers for our repositories and clearly cited it ain the manuscript, in alignment with the TC guidelines.

Kind regards

Joseph Cook and co-authors

---

## Author Response (AR4)

Joseph Cook

Aberystwyth University

Aberystwyth, Wales, UK

joc102@aber.ac.uk

**re. Minor revisions to Cook et al. Glacier algae accelerate melting of the western Greenland Ice Sheet**

Dear Editor,

The new figure has been added to the manuscript as Figure 3 and cited it in line 386:

*"Comparisons between the directional reflectance spectra gathered using the ASD field spectrometer and those measured using the UAV and Sentinel-2 are provided in Fig 3."*

Regarding the reduction of the ASD spectral data we used the reflectance at the centre wavelength of the aerial sensor and therefore did not take into account the spectral response function of the camera. We have made this utterly, explicitly clear in the manuscript at line 365 as follows:

*"Our HCRF measurements were first reduced to the reflectance values at five key wavelengths coincident with the centre wavelengths measured by the MicaSense Red-Edge camera mounted to the UAV (blue: 0.475, green: 0.560, red: 0.668, red-edge: 0.717, NIR: 0.840 μm) yielding reflectance at each wavelength as a feature vector for the classifier (in this case the spectral response function was not accounted for)."*

and later *"this process was repeated for Sentinel-2 ..."*

We have gone to great lengths across ten months to satisfy all of the reasonable requests of the reviewers and handling editor. We very much hope that this manuscript will now be accepted for publication.

Regards

Joseph Cook and co-authors